# Viral vector-mediated reprogramming of the fibroblastic tumor stroma sustains curative melanoma treatment

Sandra S. Ring[1], Jovana Cupovic[1,2], Lucas Onder[1], Mechthild Lütge [1], Christian Perez-Shibayama[1], Cristina Gil-Cruz[1], Elke Scandella[1], Angelina De Martin[1], Urs Mörbe [1], Fabienne Hartmann [1], Robert Wenger[3], Matthias Spiegl [3], Andrej Besse[4], Weldy V. Bonilla[5], Felix Stemeseder[6], Sarah Schmidt[6], Klaus K. Orlinger[6], Philippe Krebs [7], Burkhard Ludewig [1,8,11✉] & Lukas Flatz[1,9,10,11✉]

The tumor microenvironment (TME) is a complex amalgam of tumor cells, immune cells, endothelial cells and fibroblastic stromal cells (FSC). Cancer-associated fibroblasts are generally seen as tumor-promoting entity. However, it is conceivable that particular FSC populations within the TME contribute to immune-mediated tumor control. Here, we show that intratumoral treatment of mice with a recombinant lymphocytic choriomeningitis virus-based vaccine vector expressing a melanocyte differentiation antigen resulted in T cell-dependent long-term control of melanomas. Using single-cell RNA-seq analysis, we demonstrate that viral vector-mediated transduction reprogrammed and activated a *Cxcl13*-expressing FSC subset that show a pronounced immunostimulatory signature and increased expression of the inflammatory cytokine IL-33. Ablation of *Il33* gene expression in Cxcl13-Cre-positive FSCs reduces the functionality of intratumoral T cells and unleashes tumor growth. Thus, reprogramming of FSCs by a self-antigen-expressing viral vector in the TME is critical for curative melanoma treatment by locally sustaining the activity of tumor-specific T cells.

[1] Institute of Immunobiology, Kantonsspital St.Gallen, St.Gallen, Switzerland. [2] Max Planck Institute of Immunology and Epigenetics, Freiburg, Germany. [3] Department of Plastic Reconstructive Surgery, Kantonsspital St. Gallen, St. Gallen, Switzerland. [4] Department of Medical Oncology and Hematology, Kantonsspital St.Gallen, St.Gallen, Switzerland. [5] Division of Experimental Virology, Department of Biomedicine, University of Basel, Basel, Switzerland. [6] Hookipa Pharma Inc., New York, USA. [7] Institute of Pathology, University of Berne, Berne, Switzerland. [8] Institute of Experimental Immunology, University of Zurich, Zurich, Switzerland. [9] Department of Dermatology, Kantonsspital St. Gallen, St. Gallen, Switzerland. [10] Department of Dermatology, University Hospital Zurich, Zurich, Switzerland. [11] These authors jointly supervised this work: Burkhard Ludewig, Lukas Flatz. ✉email: burkhard.ludewig@kssg.ch; lukas.flatz@kssg.ch

The growth and differentiation of cancer cells in particular microenvironmental niches affects both disease progression and the clinical success of cancer therapies. The tumor microenvironment (TME) comprises various cell types including immune cells as well as non-immune cells, such as blood and lymphatic endothelial cells and fibroblastic stromal cells (FSCs)[1,2]. The presence of tumor-infiltrating CD8+ T cells, CD4+ T helper 1 cells, and CD103+ DCs is associated with improved responses to cancer immunotherapy[3–5]. Conversely, intratumoral accumulation of regulatory T cells or immunosuppressive macrophages inhibits antitumor immunity hindering the success of cancer therapy[6,7]. Likewise, tumor-associated FSCs—usually referred to as cancer-associated fibroblasts (CAFs)—have been considered as a cancer-promoting cell population that facilitates cancer progression and metastasis, e.g., through synthesis of extracellular matrix components, secretion of growth factors, cytokines, and chemokines resulting in the formation of an immunosuppressive environment[8–11]. However, tumor-associated FSC subsets can also stimulate and support antitumor immunity and thus restrain tumor growth[12–15]. Therapeutic approaches that target the fibroblastic tumor environment, such as antibodies against TGF-β, favor the development of an immune-inflamed TME and increase the efficacy of checkpoint inhibitor therapy[16–18]. Thus, it is important to elaborate therapeutic means that directly promote the remodeling of the tumor FSC landscape and thereby elicit an immune-inflamed state in the TME.

Following Coley's initial observation[19], a number of approaches have been developed to generate an inflammatory environment in the tumor (reviewed in ref. [20,21]). Oncolytic herpes viruses, for example, are designed to infect cancer cells and to elicit cancer cell apoptosis, which stimulates both presentation of tumor antigens and recruitment of immune cells to the tumor site[22]. Genetic modification of viral vectors through expression of immunostimulatory cytokines, such as colony-stimulating factor 2 (granulocyte-macrophage colony-stimulating factor) is used to overcome immunosuppressive processes in the TME through the stimulation of myeloid cells[23]. However, myeloid cell turnover in tumors is fast[24] and reprogramming of the TME via this axis might dampen sustenance of immune system-activating circuits in the TME. Hence, viral vector-mediated transduction of long-lived cells in the TME, such as FSCs that provide activating stimuli to tumor-infiltrating lymphocytes would provide a solution to the conundrum. The non-cytopathic lymphocytic choriomeningitis virus (LCMV) infects a broad range of host cells including myeloid cells, epithelial cells, endothelial cells and fibroblasts[25–29]. LCMV-based viral vectors elicit strong and multifunctional CD8+ T cell responses following intravenous application[30,31]. Clinical trials using intramuscular application of the vector have been initiated[32]. However, it is not known whether and to what extent intratumoral delivery of LCMV-based vectors impacts the TME and the outcome of the vaccination.

Here, we show that a single intratumoral application of an LCMV-based vaccine vector expressing the melanoma-associated antigen TRP2 facilitates curative melanoma treatment. Transduction of tumor-associated FSCs by the viral vector precipitates profound activation and reprogramming of immunostimulatory *Cxcl13*-expressing FSCs. We demonstrate that IL-33 catered by Cxcl13-Cre+ FSCs to tumor-infiltrating CD8+ T cells reduces local T cell exhaustion and thereby sustains control of tumor growth. In sum, our study shows that viral vector-mediated remodeling of the FSC landscape in the TME is crucial for successful tumor immunotherapy.

## Results

**Intratumoral artLCMV-TRP2 treatment controls melanoma growth.** Locally replicating LCMV strains, such as LCMV Armstrong induce robust CD8+ T cell responses that rapidly clear the virus[33]. Here, we used LCMV-based, propagation-attenuated recombinant LCMV vectors with an artificial genome organization (artLCMV)[31] expressing the murine melanocyte differentiation antigen TRP2 (artLCMV-TRP2) to treat orthotopically growing B16F10 melanomas (Fig. 1a). Therapeutic intratumoral (i.t.) treatment of established melanomas on day 7 with artLCMV-TRP2 led to tumor regression in all mice with long-term cure in ~60% of the mice, in a dose-independent manner (Fig. 1b, c, Supplementary Fig. 1a). In contrast, mice treated intravenously (i.v.) with artLCMV-TRP2 failed to eliminate the tumors, despite a transient delay in tumor growth (Fig. 1b, c). Curative i.t. treatment with artLCMV-TRP2 was dependent on self-antigen expression by the tumor cells and self-antigen-delivery by the vector, as demonstrated by the failure to eliminate TRP2-deficient tumor cells (B16F10-*Dct*−/−, Fig. 1d and Supplementary Fig. 1b) and the lack of long-term tumor control when artLCMV vectors expressing irrelevant antigens were applied (artLCMV-green fluorescent protein (GFP), Fig. 1e). Antibody-mediated T cell depletion revealed that both CD4+ and CD8+ T cell subsets were necessary to reject the tumor indicating that the antitumor effect of artLCMV-TRP2 treatment was mediated by TRP2-specific T cells (Fig. 1f and Supplementary Fig. 1c). Mice that had received curative artLCMV-TRP2 treatment were protected from re-challenge with B16F10 melanomas into the opposite flank, provided that these animals were T cell-competent (Fig. 1g). Mice cured of melanomas after i.t. artLCMV-TRP2 treatment showed vitiligo-like fur depigmentation at the tumor site (Supplementary Fig. 1d), suggesting that vector-mediated delivery of a melanocyte-specific antigen has caused a highly effective immune response. To strengthen translational relevance of our approach, we utilized a multi-metastatic model in which tumors grow simultaneously orthotopically in the skin and in the lung (Fig. 1h). Mice treated with artLCMV-TRP2 i.t. into the accessible s.c. tumor exhibited significantly reduced metastatic tumor growth in the lung compared to control mice. This finding suggests that artLCMV-induced antitumor immunity is not restricted to the injected tumor but also constrains tumor growth in peripheral tissues (Fig. 1i). In sum, these data demonstrate that i.t. artLCMV-TRP2 elicits therapeutically effective TRP2-specific T cell responses that control a metastasizing tumor and provide long-term protection from recurring melanomas.

To further elaborate the immunological mechanisms underlying the tumor-protective treatment, we followed the accumulation and functional differentiation of intratumoral T cells. Both, i.v. and i.t. artLCMV-TRP2 injection led to a significant increase of CD8+ T cell accumulation compared to PBS treatment, while CD4+ T cell infiltration was favored by the i.t. route (Fig. 2a, b and Supplementary Fig. 2a–c). Intratumoral artLCMV-TRP2 vector application not only enhanced CD8+ T cell accumulation, but significantly bolstered IFN-γ- and granzyme B-expression compared to i.v. treatment (Fig. 1c, d and Supplementary Fig. 2d). Likewise, IFN-γ production of tumor-infiltrating CD4+ T cells was significantly increased in i.t. compared to i.v. artLCMV-TRP2-treated mice (Fig. 1c and Supplementary Fig. 2e). CD8+ T cell activity in the TDLN was not affected by the vaccination route (Supplementary Fig. 2f). Overall, these data suggest that i.t. artLCMV-TRP2 injection affects mainly the TME and thereby sustains the functionality of tumor-infiltrating T cells.

**artLCMV-TRP2 targets the fibroblastic tumor stroma.** To gauge the artLCMV vector-induced alterations in the TME, we first followed the temporal and spatial dissemination of the artLCMV vectors after i.t. injection. Analysis of viral

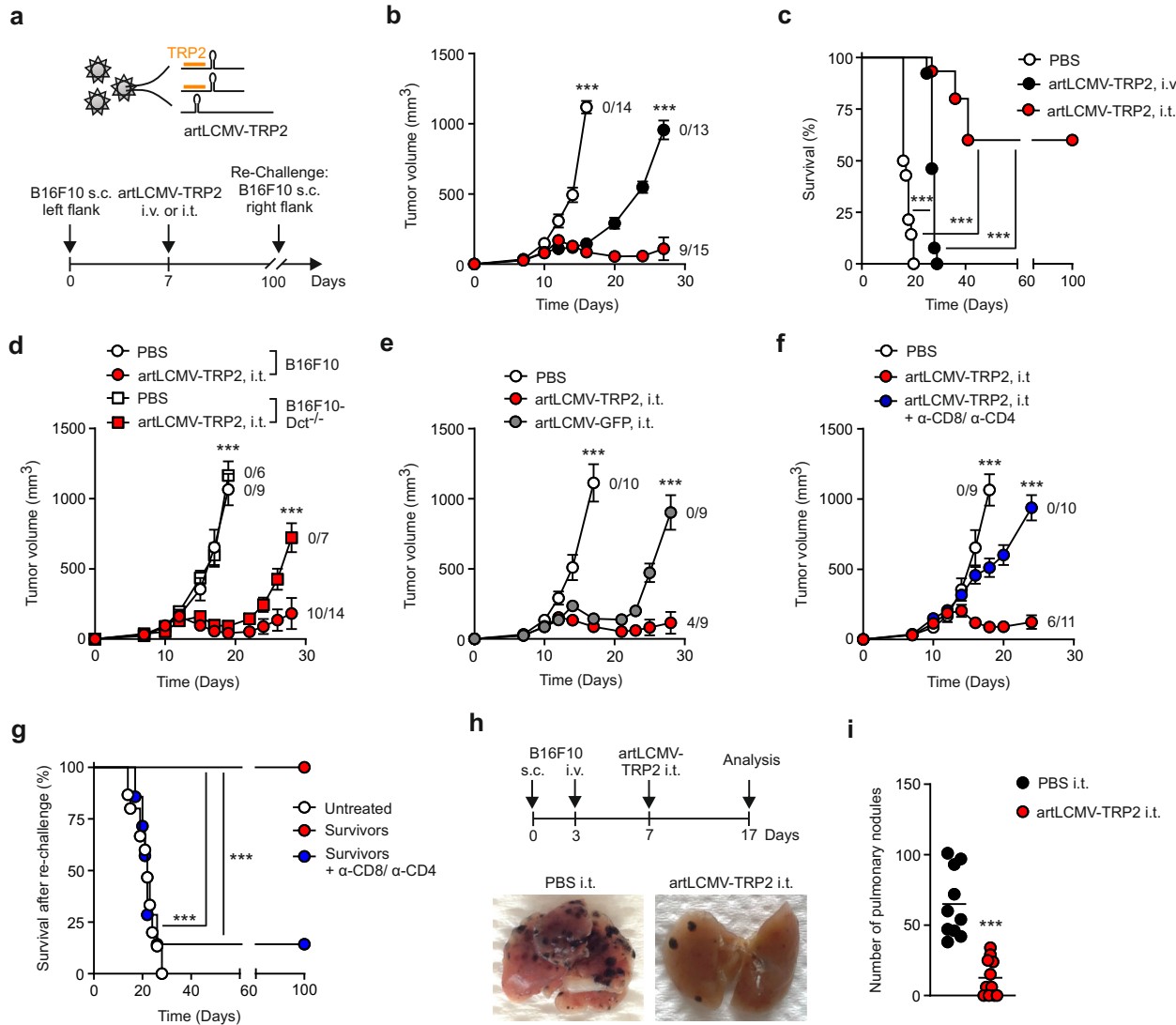

**Fig. 1 Melanoma growth after intratumoral artLCMV-TRP2 treatment. a** Schematic depiction of the recombinant artLCMV vector expressing the melanocyte differentiation antigen TRP2 (artLCMV-TRP2) and treatment scheme. **b** Tumor growth with indication of number of tumor-free mice and **c** survival of mice treated intratumorally (i.t.) or intravenously (i.v.) with artLCMV-TRP2. **d** Tumor kinetics of B16F10 and TRP2-deficient B16F10 (B16F10-$Dct^{-/-}$) in mice immunized i.t. with artLCMV-TRP2. **e** Tumor growth with indication of number of tumor-free mice of mice treated i.t. with artLCMV-TRP2 or -GFP. **f** Tumor growth in mice administered with anti-CD8- and anti-CD4-depleting antibodies and treated i.t. with artLCMV-TRP2. **g** Surviving mice from **c** were re-challenged with B16F10 s.c. into the opposite right flank. In one cohort of mice, CD8- and CD4-depleting antibodies were administered. **h, i** Mice were inoculated with s.c. B16F10 on day 0 and i.v. on day 3 to establish pulmonary melanoma metastasis. On day 7, mice were treated i.t. with artLCMV-TRP2 and on day 17 lungs were harvested. Analysis of lungs for pulmonary tumor metastasis with representative images (**h**) and quantification (**i**). Dots indicate mean ± s.e.m. values for each time point. Pooled data from three independent experiments with $n = 14$ (PBS), $n = 13$ (artLCMV-TRP2, i.v.) and $n = 15$ (artLCMV-TRP2, i.t.) mice (**b, c**), $n = 15$ (Untreated), $n = 13$ (Survivors) and $n = 7$ (Survivors + α-CD8/α-CD4) mice (**g**); $n = 10$ (PBS, i.t.) and $n = 11$ (artLCMV-TRP2, i.t.) mice (**i**). Pooled data from two independent experiments with $n = 9$ (B16F10, PBS), $n = 6$ (B16F10-Dct$^{-/-}$, PBS), $n = 14$ (B16F10, artLCMV-TRP2, i.t.) and $n = 7$ (B16F10-Dct$^{-/-}$, artLCMV-TRP2, i.t.) mice (**d**); $n = 10$ (PBS), $n = 9$ (artLCMV-GFP, i.t.) and $n = 9$ (artLCMV-TRP2, i.t.) mice (**e**); $n = 9$ (PBS), $n = 11$ (artLCMV-TRP2, i.t.) and $n = 10$ (artLCMV-TRP2, i.t.+α-CD8/α-CD4) mice (**f**). Statistical analysis was performed using two-way analysis of variance (ANOVA) with Bonferroni (**b, d, e, f**), Log-Rank Mantel–Cox test (**c, g**), or unpaired two-tailed Student's t test (**i**) with *P < 0.05: **P < 0.01; ***P < 0.001. Source data and exact P values are provided in the Source data file.

nucleoprotein (LCMV-NP) RNA expression revealed the persistence of the vector in the tumor tissue with minor spread to the tumor-draining lymph node (TDLN), remote non-draining lymph nodes or the spleen (Fig. 3a). Flow cytometric analysis of single-cell suspensions at day 11 showed that the LCMV-NP protein was broadly distributed with high expression in PDPN-expressing FSCs and in B16F10 tumor cells (marked by the mCherry reporter), while a significantly lower proportion of CD45$^+$ immune cells and CD31$^+$ blood endothelial cells harbored the viral antigen (Fig. 3b, c, Supplementary Fig. 3a, b). In

comparison, i.v. artLCMV vector application led only to low or moderate transduction of cells in the TME (Supplementary Fig. 3b, c). Confocal microscopy analysis confirmed the presence of LCMV-NP in melanoma cells (Fig. 3d, arrows) and in PDPN$^+$ FSCs located mainly at the tumor margin (Fig. 3d, arrowheads). FSCs isolated from human skin and melanoma biopsies could be readily infected with artLCMV vaccine vectors (Fig. 3e, f) with the majority of LCMV-NP$^+$ FSCs showing co-expression of CD90 (Thy1) and PDPN (Fig. 3g, Supplementary Fig. 3d, e). In contrast, patient-matched PBMCs showed a significantly lower

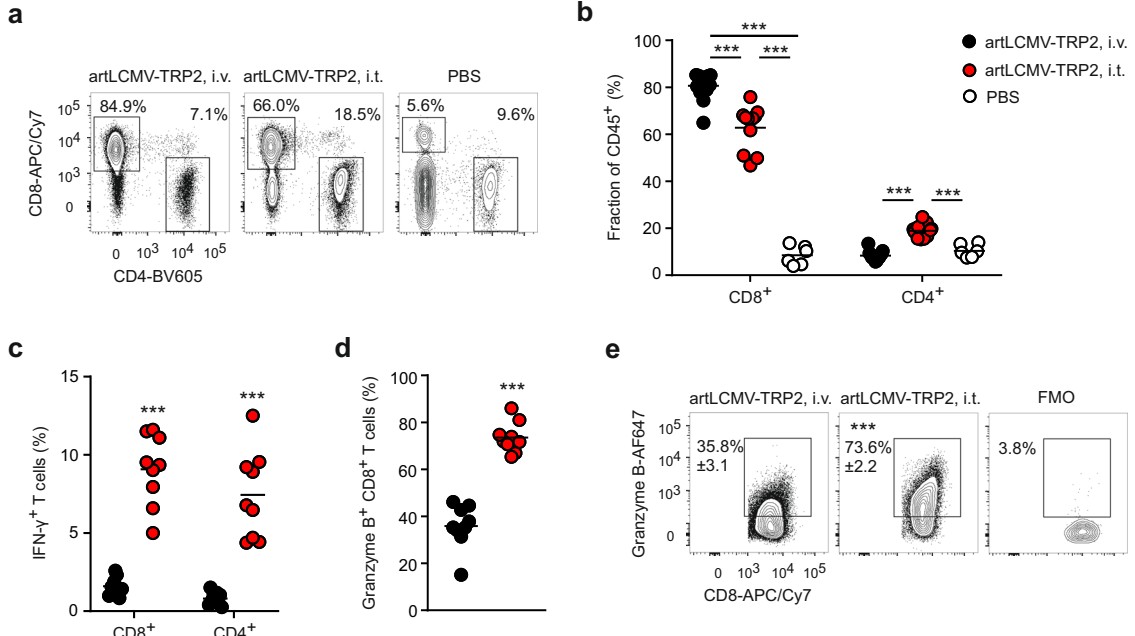

**Fig. 2 T cell activity after intratumoral artLCMV-TRP2 treatment. a–d** B16F10-tumor bearing mice were treated on day 7 with artLCMV-TRP2 i.v. or i.t. Flow cytometric analysis of tumor-infiltrating T cells on day 15: Representative plots (**a**) and quantification of CD8$^+$ and CD4$^+$ T cell frequency (**b**). **c** IFN-γ-producing CD8$^+$ and CD4$^+$ T cells. Granzyme B-expressing CD8$^+$ T cells (**d**) with representative plots (**e**). Dots represent individual mice and lines indicate mean values. Pooled data from two independent experiments with $n = 6$ (PBS), $n = 13$ (artLCMV-TRP2, i.v.) and $n = 11$ (artLCMV-TRP2, i.t.) mice (**b**); $n = 9$ (artLCMV-TRP2, i.v. and i.t.) mice (**c, d**). Statistical analysis was performed using one-way ANOVA with Tukey's multiple comparison test (**b**) or unpaired two-tailed Student's $t$ test (**c, d**) with *$P < 0.05$: **$P < 0.01$; ***$P < 0.001$. Source data and exact $P$ values are provided in the Source data file.

transduction rate (Fig. 3f) with mainly myeloid cells harboring the viral protein (Supplementary Fig. 3g, h). These data indicate that both murine and human melanoma-derived FSCs can be targeted efficiently by the artLCMV vector.

Intratumoral application of artLCMV vectors changed the non-hematopoietic tumor stroma with a significantly increased proportion of PDPN$^+$ FSCs (Fig. 4a), while the fractions of CD31$^+$ blood endothelial cells and PDPN-negative FSCs cells were reduced (Supplementary Fig. 4a–c). The stromal cell composition in the TDLN was not affected by the i.t. artLCMV vector treatment (Supplementary Fig. 4b, d). PDPN-expressing tumor FSCs were activated by the artLCMV injection with significantly enhanced expression of MHC class I molecules (H2-K$^b$ and D$^b$), intercellular adhesion molecule 1 and vascular cell adhesion molecule 1 (Supplementary Fig. 4e). Moreover, intratumoral artLCMV application induced a significant increase in gene expression of lymphocyte-attracting chemokines *Cxcl13* and *Ccl19* in PDPN$^+$ FSCs (Fig. 4b). The remodeling of the FSC landscape with the acquisition of immunostimulatory properties was confirmed in Cxcl13-Cre/tdTomato R26R-EYFP (abbreviated as Cxcl13-Cre/tdTom EYFP) mice. In this transgenic mouse model, Cre recombinase expression is controlled by the *Cxcl13* promoter and facilitates labeling of cells with current or past *Cxcl13* expression or the progeny of chemokine-expressing cells with EYFP. In addition, expression of the real-time reporter tdTomato is under the control of the *Cxcl13* promoter allowing detection of current *Cxcl13* expression mice[34,35]. Microscopic analysis of B16F10 tumors growing in Cxcl13-Cre/tdTomato EYFP mice revealed that lineage-traced EYFP$^+$ cells localized mainly in the tumor margin co-express PDPN both in PBS- and artLCMV-treated tumors (Fig. 4c, arrowheads). Flow cytometry-based quantification of tumor fibroblasts showed that artLCMV-TRP2 treatment precipitated a significant expansion of PDPN$^+$ EYFP$^+$ FSCs (Fig. 4d), while the proportion of EYFP$^+$ PDPN$^+$

FSCs in TDLNs remained unchanged (Supplementary Fig. 4f). Importantly, current expression of CXCL13 as determined by the tdTomato reporter was significantly increased in tumor fibroblasts following artLCMV-TRP2 treatment, but was not affected in TDLNs (Fig. 4e, Supplementary Fig. 4g). In sum, these data indicate that the transduction of TME-associated PDPN$^+$ FSCs by artLCMV-TRP2 promotes the activation of immunostimulatory, *Cxcl13*-expressing PDPN$^+$ FSCs.

**artLCMV-TRP2 application reprograms the fibroblastic tumor stroma.** To resolve the molecular circuits underlying the artLCMV-induced activation of FSCs in the TME, we isolated FSCs from tumors of Cxcl13-Cre/tdTom EYFP mice and performed droplet-based single-cell RNA sequencing (scRNA-seq; Supplementary Fig. 5a). Unsupervised clustering of EYFP-expressing cells using Uniform Manifold Approximation and Projection (UMAP) revealed six conserved FSC clusters (Fig. 5a). Based on hierarchical clustering with computation of cluster-specific genes (Supplementary Fig. 5b) and the expression of canonical FSC markers (Supplementary Fig. 5c), we discerned transcriptional signatures associated with two clusters of inflammatory cancer-associated fibroblasts (iCAF1 and iCAF2) expressing *Dpp4, Cd34, Ly6c1*, and *Ly6a* (Fig. 5b). These particular populations phenotypically resemble iCAFs present in murine and human pancreatic ductal adenocarcinoma[13,36]. Moreover, we distinguished two clusters of myofibroblasts (myCAF1 and myCAF2) characterized by high expression of *Col15a1, Tgfb1, Lrrc15*, and the MHC II molecule *H2-Ab1* in myCAF2 (Supplementary Fig. 5d). Expression of ECM genes in the myCAF1 population suggests a more structural role through the production of collagen networks. MyCAFs2 express potential immune-regulatory genes as well as genes associated with hypoxia and metabolic regulatory pathways, recently described as VEGF$^+$ CAFs[18]. Moreover, we found a cluster of mural cells marked by

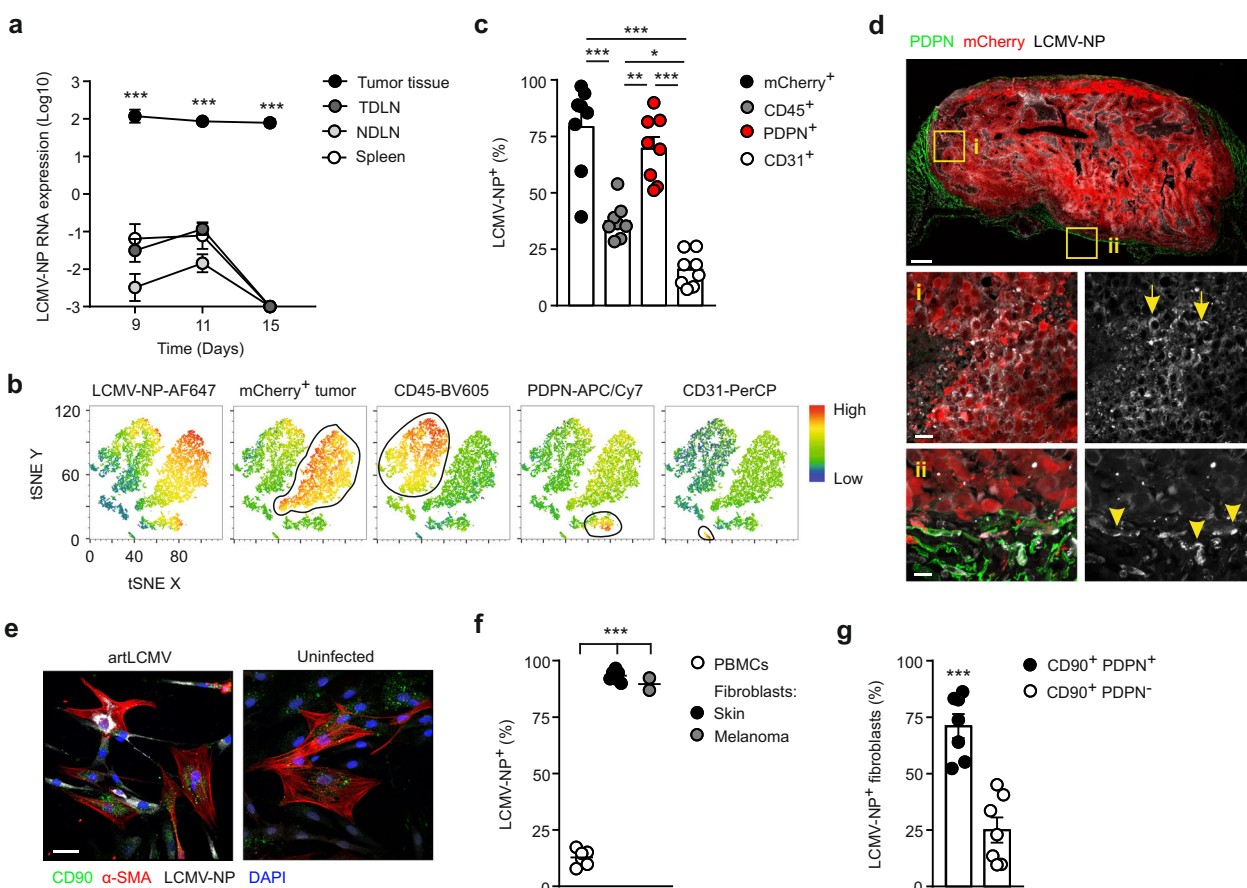

**Fig. 3 Transduction of PDPN⁺ FSCs in the TME by intratumoral artLCMV-TRP2 application.** Mice were inoculated with B16F10-mCherry and immunized i.t. on day 7 with artLCMV-TRP2. **a** LCMV-NP RNA expression in tumor, tumor-draining, and non-draining lymph nodes (TDLN and NDLN) and spleen at indicated time points after i.t. artLCMV-TRP2 treatment. Dots indicate mean±s.e.m. for each time point. **b, c** Flow cytometric analysis with representative tSNE plot gated on viable Ter119⁻ cells on day 11. **b** Expression of indicated marker and **c** frequency of LCMV-NP⁺ cells among identified cell populations in the TME including: mCherry⁺ tumor cells, CD45⁺ immune cells, PDPN⁺ FSCs, and CD31⁺ BECs. Dots represent individual mice and lines indicate mean values ± s.e.m values. **d** Representative confocal microscopy of tumors on day 11 following i.t. artLCMV-TRP2 injection from n = 3 mice. Representative areas are shown: (i) in the center of the tumor and (ii) in the tumor margin. Arrows and arrowheads indicate artLCMV-transduced cells by positive staining for LCMV-NP. Scale bar 1000 μm (overview) and 10 μm (boxed areas). **e–g** Infection of FSCs from human skin and melanoma biopsies with artLCMV. Scale bar 50 μm. **e** Representative high resolution immunofluorescence images of artLCMV-transduced human skin-derived FSCs. **f** Frequency of LCMV-NP⁺ cells. **g** Frequency of CD90⁺ PDPN⁺ and CD90⁺ PDPN⁻ cells among LCMV-NP⁺ FSCs. Dots represent individuals and lines indicate mean values ± s.e.m values. Pooled data from two independent experiments with n = 11 mice (**a**) and n = 8 mice (**c**). Pooled data from n = 6 (PBMCs and Skin Fibroblasts) and n = 2 (Melanoma Fibroblasts) patients (**f**) and n = 7 patients (**g**). Statistical analysis was performed using two-way analysis of variance (ANOVA) with Bonferroni (**a**), one-way ANOVA with Tukey's multiple comparison test (**c**), or unpaired two-tailed Student's *t* test (**f, g**) with *P < 0.05; **P < 0.01; ***P < 0.001. Source data and exact P values are provided in the Source data file.

*Rgs5* and *Des* (Supplementary Fig. 5e) and a cluster of fibroblasts enriched in cell cycle genes including *Mki67*, *Cks2*, and *Cdc20* (proliferating CAFs, Supplementary Fig. 5b, f). Detailed analysis of EYFP⁺ FSCs unveiled a substantial shift in the abundance of the iCAF1 and iCAF2 clusters in artLCMV-treated tumors, whereas the relative abundance of the other clusters was largely preserved (Fig. 5c). Flow cytometric analysis for CD26 (*Dpp4*) and CD34 expression confirmed that iCAFs represent a substantial fraction of the EYFP⁺ PDPN⁺ tumor fibroblasts and that the subsets of Ly6C⁺ CD34⁺ and Sca-1⁺ CD34⁺ fibroblasts were significantly expanded after artLCMV treatment (Fig. 5d and Supplementary Fig. 5g).

Computation of differential gene expression for the iCAF clusters under artLCMV treatment vs. PBS control revealed distinct sets of immunostimulatory pathways (Supplementary Fig. 6a). Moreover, we found an iCAF signature (Supplementary Fig. 6b) that was particularly strong in the iCAF2 cluster as revealed by the alignment of the individual cells of all clusters

along their trajectories in diffusion maps (Fig. 5e). The distinct immunostimulatory gene signatures, i.e., cell activation, chemokine and cytokine expression (Fig. 5f), antigen presentation, and type I interferon, and interferon-stimulated genes (Supplementary Fig. 6c), mapped preferentially to the iCAF subsets. The expression of the chemokine gene *Cxcl13* was almost exclusively restricted to FSCs in the iCAF2 cluster both in PBS- and artLCMV-injected tumors (Fig. 5g). TdTomato⁺ cells in tumors of artLCMV-TRP2-treated Cxcl13-Cre/tdTom EYFP mice belonged mainly to the Ly6C⁺ CD34⁺ and Sca-1⁺ CD34⁺ subsets, which is compatible with the iCAF signature (Supplementary Fig. 6d). Global analysis of differentially regulated genes under artLCMV- vs. PBS-treatment in the iCAF clusters highlighted *Il33* as a potential candidate for the local stimulation of T cells (Fig. 5h). The increased expression of *Il33* in EYFP⁺ PDPN⁺ FSCs following artLCMV application was validated by real-time PCR analysis (Fig. 5i). Moreover, microscopic analysis of IL-33-expressing cells in the TME of artLCMV-TRP2-treated

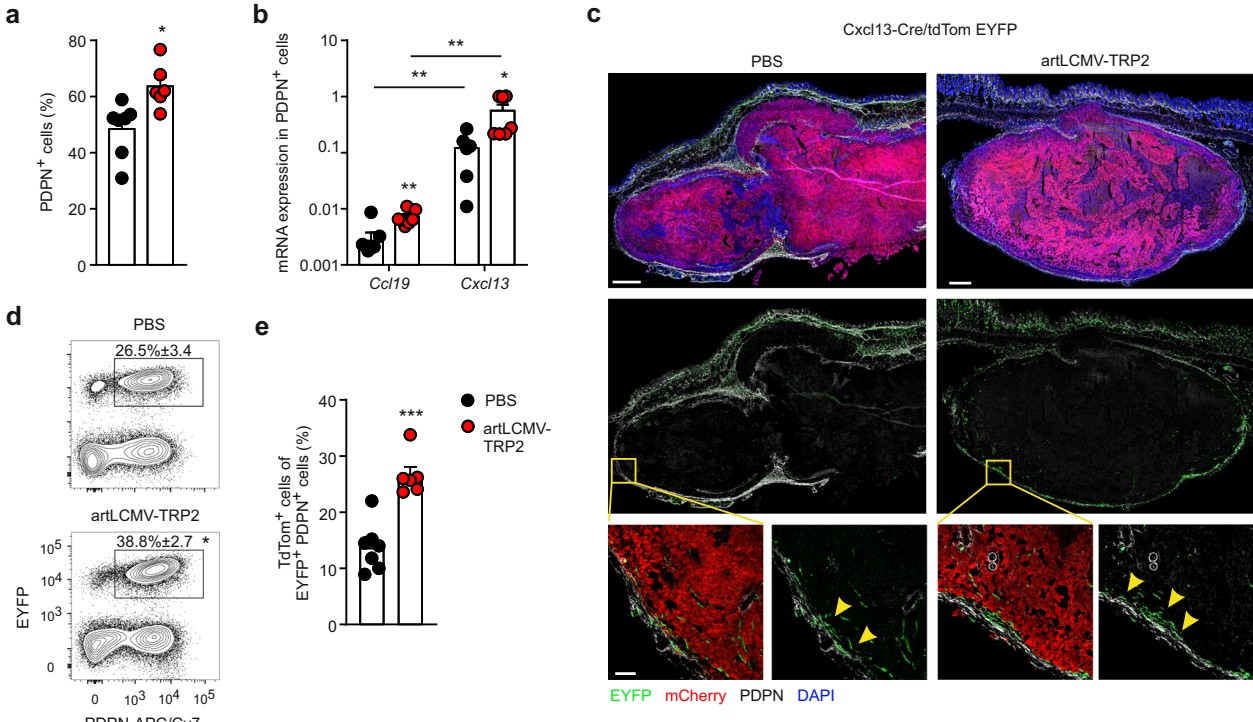

**Fig. 4 artLCMV treatment induces *Cxcl13*-expressing FSCs in the TME.** Mice were inoculated with B16F10-mCherry and immunized i.t. on day 7 with artLCMV-TRP2 and tumors were analyzed on day 11. **a** Frequency of PDPN+ cells in tumors from mice treated i.t. with artLCMV-TRP2. **b** Quantitative real-time PCR for *Ccl19* and *Cxcl13* mRNA expression in PDPN+ FSCs sorted from tumors on day 11. **c** Confocal microscopy of tumors from Cxcl13-Cre/tdTom EYFP mice treated as indicated. Scale bars 700 μm (overview) and 80 μm (boxed areas). Arrowheads indicate EYFP+ PDPN+ FSCs. **d** Frequency of EYFP+ PDPN+ in Cxcl13-Cre/tdTom EYFP mice. **e** Frequency of tdTom+ cells among EYFP+ PDPN+ cells as marker for current CXCL13 expression. Dots represent individual mice and lines indicate mean values ± s.e.m values. Pooled data from two independent experiments with n = 7 (PBS) and n = 6 (artLCMV-TRP2) mice (**a**, **d**, **e**); n = 7 (*Ccl19*, PBS; *Ccl19*, artLCMV-TRP2; *Cxcl13*, artLCMV-TRP2) and n = 6 (*Cxcl13*, PBS) mice (**b**). Representative images from n = 4 (PBS) and n = 5 (artLCMV-TRP2) mice (**c**). Statistical analysis was performed using unpaired two-tailed Student's *t* test (**a**, **b**, **d**, **e**) with *P < 0.05: **P < 0.01; ***P < 0.001. Source data and exact P values are provided in the Source data file.

Cxcl13-Cre/tdTom EYFP mice revealed cytokine-positive EYFP+ cells at the tumor margin (Supplementary Fig. 6e). In sum, these data indicate that intratumoral artLCMV application reprograms the tumor-associated FSC landscape leading to a pronounced shift towards an immunostimulatory cell state in iCAFs resulting in a higher abundance of the iCAF2 subset.

**FSC-derived IL-33 prevents intratumoral T cell exhaustion.** Since tumor FSCs exhibited increased IL-33 production after artLCMV-TRP2 injection, we considered it likely that CD8+ T cells respond locally to this cytokine. Indeed, tumor-infiltrating CD8+ T cells showed elevated expression of the IL-33 receptor ST2, while ST2 expression on CD8+ T cells in the TDLN remained low (Fig. 6a and Supplementary Fig. 7a). To further assess to which extent FSC-derived IL-33 in the TME affects treatment efficacy, we crossed Cxcl13-Cre/tdTom mice to *Il33*fl/fl (Cxcl13-Cre *Il33*fl/fl) mice. Tumor growth and survival were similar in PBS-injected Cre-negative littermate Ctrls and Cxcl13-Cre *Il33*fl/fl mice. Moreover, while intratumoral artLCMV-TRP2 treatment of Ctrl mice promoted tumor control and cured four out of nine mice in long-term, Cxcl13-Cre *Il33*fl/fl mice failed to control the tumors (Fig. 6b, c). Analysis of the tumoral immune cell milieu revealed decreased frequencies of tumor-infiltrating CD8+ T cells on day 15 after i.t. artLCMV-TRP2 treatment in Cxcl13-Cre *Il33*fl/fl compared with Ctrl mice, whereas the abundance of other immune cell subsets was not substantially affected (Fig. 6d and Supplementary Fig. 7b, d). Notably, the frequency of CD8+ and CD4+ T cells in the TDLN was not altered due to the *Il33*-

deficiency in Cxcl13-Cre+ FSCs (Supplementary Fig. 7e, f). Further flow cytometric analyses revealed decreased frequencies of TRP2-specific, tetramer-binding CD8+ T cells in tumors from Cxcl13-Cre *Il33*fl/fl mice (Fig. 6e), a reduction of KLRG1+ CD62L− CD8+ T cells (Fig. 6f and Supplementary Fig. 7g), and significantly impaired CD8+ T cell effector function as demonstrated by the reduced expression of IFN-γ and TNF-α following ex vivo restimulation with TRP2 peptide (Fig. 6g). In addition to the loss of CD8+ effector T cells, the lack of *Il33* expression in Cxcl13-Cre+ FSC in the TME led to an increase of exhausted CD8+ T cells with higher abundance of CD69-expressing cells (Fig. 6h and Supplementary Fig. 7h), elevated expression of the exhaustion markers PD-1, Eomes, TOX, but not T-bet, in CD8+ T cells (Fig. 6i), increased abundance of PD-1high Eomeshigh CD8+ T cells (Fig. 6j and Supplementary Fig. 7i) and PD-1high TOXhigh CD8+ T cells (Fig. 6k and Supplementary Fig. 7j). Importantly, we did not find differences in the expression of CD8+ T cells exhaustion markers in the TDLN between Ctrl and Cxcl13-Cre *Il33*fl/fl mice (Supplementary Fig. 7k). Moreover, the production of effector cytokines in adoptively transferred P14 T cells (T cell receptor transgenic CD8+ T cells specific for the LCMV epitope GP33-41) was significantly decreased by *Il33*-ablation in Cxcl13-Cre+ tumor FSCs and deficiency of the IL-33 receptor ST2 on transferred P14 T cells (Supplementary Fig. 8a–c).

To further assess whether local application of the LCMV vector impacts T cell responses in the draining lymph node, we infected Cxcl13-Cre *Il33*fl/fl and Cre-negative Ctrl mice subcutaneously with artLCMV-TRP2 (Supplementary Fig. 9a). Analysis of the skin-draining lymph node (SDLN) on day 8 post immunization

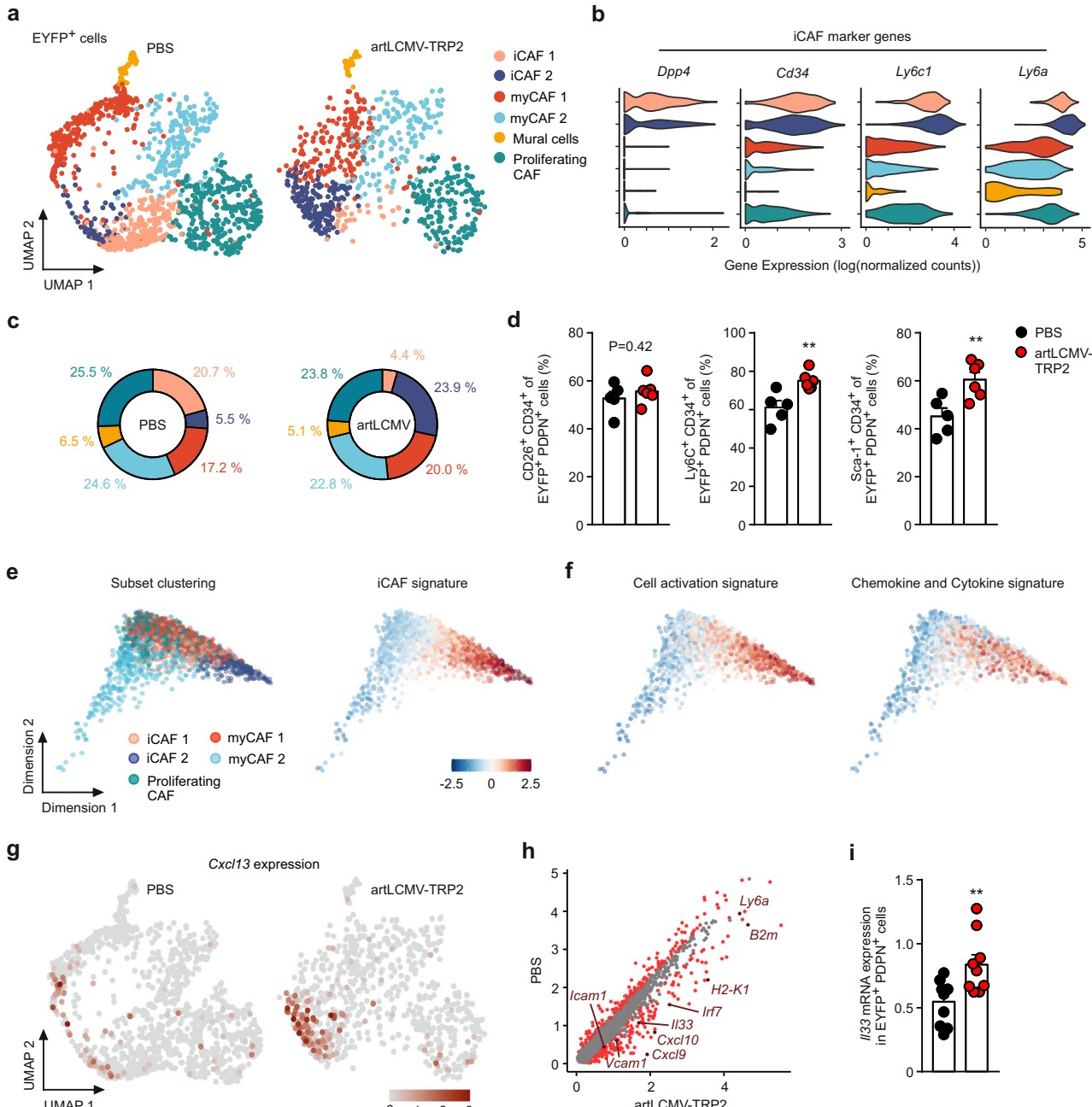

**Fig. 5 Activation of *Cxcl13*-expressing immune-stimulatory FSCs by intratumoral artLCMV-TRP2 treatment.** Single-cell RNA-seq analysis of EYFP+ cells isolated from the tumors of Cxcl13-Cre/ tdTom EYFP mice treated i.t. with artLCMV-TRP2 or PBS on day 11. **a** UMAP plots displaying FSCs cluster assignment in PBS control or artLCMV-treated mice. **b** Violin plots depicting expression of iCAF marker genes *Ly6c1, Ly6a, Cd34*, and *Dpp4*. **c** Pie charts showing the relative abundance of the identified FSC clusters. **d** Flow cytometric analysis of EYFP+ PDPN+ cells stained for CD26 (Dpp4), CD34, Ly6C (Ly6c1), and Sca-1 (Ly6a/ Ly6e). **e, f** Diffusion maps of EYFP+ tumor FSCs with trajectories constructed based on differential genes analysis in clusters iCAF1 and iCAF2 between PBS and artLCMV condition. **e** Diffusion maps showing subset clustering and iCAF signatures and **f** cell activation signature and chemokine and cytokine signature. **g** UMAP feature plots of *Cxcl13* expression. **h** Scatter plot depicting DE genes of iCAF1 and iCAF2 between PBS control and artLCMV-TRP2-treated mice. **i** Real-time PCR analysis for *Il33* expression in EYFP+ PDPNhi sorted cells from tumors of Cxcl13-Cre/tdTom EYFP mice. ScRNA-seq analysis was performed with one biological replicate for $n = 6$ PBS-treated mice and two biological replicates for $n = 3–4$ artLCMV-TRP2-treated mice. We obtained 1293 (PBS) and 786 (artLCMV-TRP2) EYFP-expressing cells. Dots in **d, i** represent individual mice and shown is mean + s.e.m. values. Pooled data from two independent experiments with $n = 5$ (PBS) and $n = 6$ (artLCMV-TRP2) mice (**d**); $n = 10$ (PBS) and $n = 9$ (artLCMV-TRP2) mice (**i**). Statistical analysis was performed using unpaired two-tailed Student's *t* test (**d, i**) with *$P < 0.05$: **$P < 0.01$; ***$P < 0.001$. Source data and exact *P* values are provided in the Source data file.

did not reveal significant differences in the frequency of CD8+ T cells (Supplementary Fig. 9b), or TRP2- and LCMV (GP33 and NP396)-specific CD8+ T cell cytokine responses between Cxcl13-Cre *Il33*fl/fl and Cre-negative Ctrl mice (Supplementary Fig. 9b, f).

These data indicate that IL-33 is dispensable for the activation of CD8+ T cell effector function in the SDLN after local artLCMV-TRP2 immunization. Collectively, we conclude that IL-33 catered by Cxcl13-Cre+ FSCs to CD8+ T cells in the TME sustains T cell

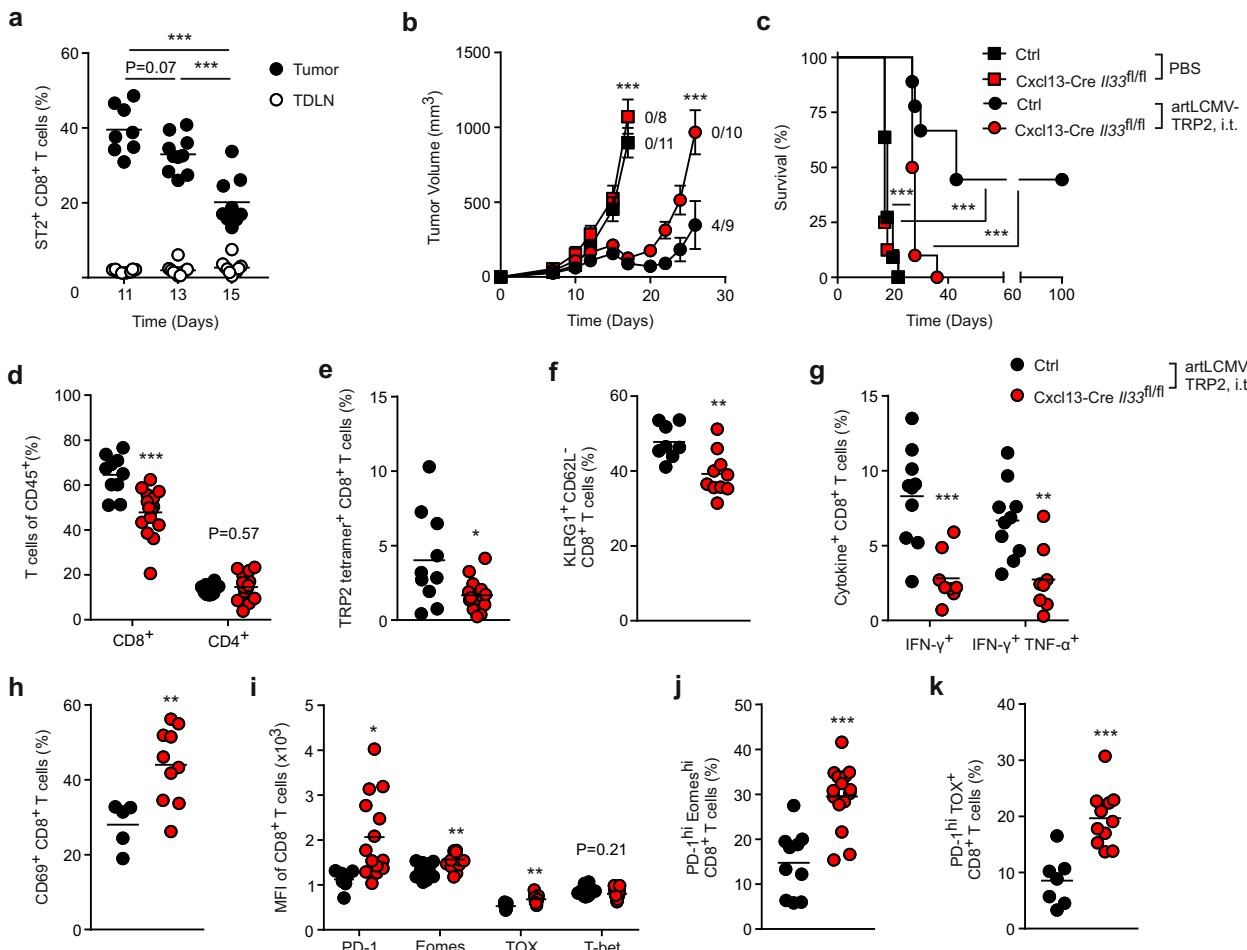

**Fig. 6 Sustenance of antitumor T cell responses by FSC-derived IL-33. a** Mice were inoculated s.c. with B16F10 melanoma cells and treated on day 7 i.t. with artLCMV-TRP2. Frequency of ST2+ CD8+ T cells in the tumor and the TDLN at indicated time points. **b** Tumor growth kinetics and **c** survival in Cxcl13-Cre *Il33*fl/fl and Cre-negative littermate mice (Ctrl) treated i.t. with artLCMV-TRP2 on day 7 after B16F10 inoculation. Dots indicate mean ± s.e.m. for each time point. **d**–**k** Tumor-infiltrating T cells were analyzed by flow cytometry on day 15. **d** Frequency of tumor-infiltrating CD8+ and CD4+ T cells. **e** Frequency of TRP2 tetramer+ CD8+ T cells and **f** KLRG1+ CD62L− CD8+ T cells. **g** IFN-γ- and TNF-α-producing tumor-infiltrating TRP2-specific CD8+ T cells. **h** Frequency of CD69+ CD8+ T cells. **i** Mean expression of Eomes, PD-1, TOX, and T-bet on tumor-infiltrating CD8+ T cells and **j** Frequency of PD-1hi Eomeshi and **k** PD-1hi TOX+ among CD8+ T cells. Dots represent individual mice and lines indicate mean values. Pooled data from two independent experiments with n = 8 (Day 11), n = 10 (Day 13) and n = 9 (Day 15) (**a**); n = 11 (PBS, Ctrl), n = 8 (PBS, Cxcl13-Cre *Il33*fl/fl), n = 9 (artLCMV-TRP2, i.t., Ctrl) and n = 10 (artLCMV-TRP2, i.t., Cxcl13-Cre *Il33*fl/fl) mice (**b**, **c**); n = 8 (Ctrl) and n = 10 (Cxcl13-Cre *Il33*fl/fl) mice (**f**); n = 10 (Ctrl) and n = 8 (Cxcl13-Cre *Il33*fl/fl) mice (**g**); n = 5 (Ctrl) and n = 10 (Cxcl13-Cre *Il33*fl/fl) mice (**h**); n = 7 (TOX, Ctrl) and n = 9 (TOX, Cxcl13-Cre *Il33*fl/fl) mice (**i**); n = 7 (Ctrl) and n = 11 (Cxcl13-Cre *Il33*fl/fl) mice (**k**). Pooled data from three independent experiments with n = 10 (Ctrl) and n = 15 (Cxcl13-Cre *Il33*fl/fl) mice (**d**, **e**, **j**); n = 7 (PD-1, Ctrl), n = 9 (T-bet, Ctrl), n = 10 (Eomes, Ctrl) and n = 9 (PD-1, Eomes, T-bet, Cxcl13-Cre *Il33*fl/fl) mice (**i**). Statistical analysis was conducted using one-way ANOVA with Tukey's multiple comparison test (**a**), two-way analysis of variance (ANOVA) with Bonferroni (**b**), Log-Rank Mantel–Cox test (**c**) or unpaired two-tailed Student's *t* test (**d**–**k**) with *P < 0.05; **P < 0.01; ***P < 0.001. Source data and exact P values are provided in the Source data file.

effector differentiation and prevents T cell exhaustion, hence enabling curative melanoma treatment by LCMV-based viral vectors.

## Discussion

This study unveils an inducible intratumoral fibroblastic niche that fosters the activity of antitumor CD8+ T cells. Single intratumoral administration of a propagation-attenuated artLCMV vector profoundly modulated the activation state of immunostimulatory, *Cxcl13*-expressing FSCs in the TME. IL-33 was identified as a key factor provided locally by reprogrammed iCAFs that fuels the activity of antitumor T cells and thereby facilitates curative melanoma treatment.

Viral vectors are considered a promising means to activate and mobilize antitumor T cells[21,37]. Oncolytic viruses, for example,

are used for cancer therapy because they directly lyse infected tumor cells and subsequently induce an inflammatory immune response[38]. Several disadvantages hinder the long-term efficacy of oncolytic viruses in cancer therapy including type-I IFN-constrained replication of oncolytic viruses within tumor cells[39], limited selective targeting of tumor cells, or temporally restricted viral persistence due to lysis of infected cancer cells[40]. In contrast, LCMV is a highly immunogenic but non-cytopathic virus that persists in long-lived host cells unless these cells are lysed by CD8+ T cells[41]. Molecular modification of the artLCMV genome[30] allows for the incorporation of diverse tumor-derived antigens, such as the melanoma-associated TRP2 antigen, to induce tumor-specific T cell responses. Moreover, our study shows that artLCMV-transduction of tumor-associated FSCs results in a long-lasting persistence of vector-encoded antigens

and that the vector-induced inflammatory milieu in the TME controls the maintenance of antitumor immunity and tumor regression.

Recent high-dimensional transcriptomic analyses have begun to unravel the heterogeneity of the fibroblastic landscape in healthy tissue and in different human and murine tumors[13,36,42,43]. Our study confirms the existence of iCAFs and myCAFs as two major tumor FSC subsets that are similar to immune and desmoplastic CAFs[42] and VEGF+ CAFs[18]. While EYFP-labeled FSCs in Cxcl13-Cre/tdTom EYFP mice comprised both iCAFs and myCAFs, current *Cxcl13* mRNA expression was almost exclusively restricted to iCAF2 in artLCMV-injected tumors. Thus, the Cxcl13-Cre/tdTom EYFP mouse model facilitates the phenotypical characterization of immune-stimulatory *Cxcl13*-expressing iCAFs in the TME and enabled the resolution of the molecular circuits underlying the artLCMV-induced activation of tumor FSCs.

Our study delineates an intervention strategy to modulate both the global composition of the fibroblastic TME and the functional state of those fibroblasts that form immunostimulatory niches in melanoma. It is mainly the iCAF population that is reprogrammed by LCMV-based vectors to attract and sustain tumor-infiltrating CD8+ T cells through: (1) increased intratumoral accumulation of CD8+ T cells via type I IFN-induced chemokines; (2) increased antigen presentation; and (3) prevention of CD8+ T cell exhaustion by FSC-derived molecules, such as IL-33. Moreover, the expression of the lymphocyte-attracting chemokines *Cxcl13* and *Ccl19* in iCAF2 was significantly increased following intratumoral artLCMV application, which might favor the development of tertiary lymphoid structures and could improve the efficacy of checkpoint inhibitor therapy[44–46]. Hence, application of artLCMV should be considered as a treatment option to harness the immunostimulatory capacities of reprogrammed iCAFs in cancer entities that are amenable to checkpoint inhibitor therapy and are accessible for intratumoral delivery of the viral vector.

Fibroblasts are the most abundant cellular source of IL-33 in many tissues including the skin and the lung[47,48]. In addition to the dominant function of IL-33 in the induction of type-2 immunity, recent studies have assigned an important role of this tissue cytokine during antiviral and antitumoral immune responses[49–51]. Our finding that mice with *Il33*-deficiency in Cxcl13-Cre+ FSCs exhibit impaired effector function of TRP2-specific CD8+ T cells are in line with previous studies that showed IL-33-dependent T cell effector differentiation[27,52]. The increased expression of the exhaustion markers PD-1, Eomes, and TOX on tumor-infiltrating CD8+ T cells in mice lacking intratumoral IL-33 production by skin-derived Cxcl13-Cre+ FSCs is consistent with the phenotype of terminally exhausted T cells[53–55]. Hence, distinct FSC subsets in the melanoma TME provide niches that support the functionality of antitumor T cells and prevent exhaustive activation. Similar T cell activity-supporting niches exist in other organs, such as the lung, where perivascular FSCs support not only type 2 innate lymphoid cells[47], but sustain inflating memory CD8+ T cells in an IL-33-dependent manner[56]. Thus, it appears that artLCMV-mediated reprogramming of the fibroblastic TME triggers immune cell-activating circuits that support local T cell responses in virally infected peripheral tissues.

Overall, the present study provides evidence that immune responses elicited by local artLCMV-TRP2 treatment are not limited to the accessible injected tumor site but extend systemically as shown by the significant reduction of tumor growth in the lung. Thus, reprogramming of the local tumor-associated fibroblastic stroma by artLCMV bolsters effector functions of tumor-

specific T cells that traffic to distant metastatic lesions to exert systemic antitumor activity. Currently, artLCMV vectors are in clinical trials for the vaccination of transplant patients against cytomegalovirus infections[32] (ClinicalTrials.gov Identifier NCT03629080) and for the treatment of patients with human papilloma virus-induced head and neck cancer (ClinicalTrials.gov Identifier NCT0418021). Our study provides the rationale for the intratumoral application of artLCMV vectors for other tumor entities including melanoma. It will be important in future studies to further elaborate whether and to which extent artLCMV-based intratumoral vaccination can be combined with established therapeutical approaches to foster and sustain antitumor T cell activity in the inflamed TME and in distant metastatic lesions.

## Methods

**Mice**. C57BL/6N (B6) mice were obtained from Charles River Laboratories (Germany). Cxcl13-Cre/tdTomato crossed to R26R-EYFP reporter mice have been previously described[34,35]. The *Il33*-targeting vector to generate the *Il33*[fl/fl] mice was generated by the trans-NIH Knock-Out Mouse Project (KOMP Project ID CSD88909) and obtained from the KOMP Repository (www.komp.org). To specifically ablate *Il33* expression in FSCs, Cxcl13-Cre/tdTomato mice were crossed with *Il33*[fl/fl] mice. Transgenic mice expressing the P14 T cell receptor specific for the H-2D[b]-restricted epitope GP$_{33-41}$ and CD45.1 as congenic marker were used as organ donors for adoptive CD8+ T cell transfer experiments[57]. P14 TCR transgenic mice crossed to *Il1rl1* (ST2)-deficient mice. *Il1rl1*[−/−] P14 mice were used under MTA from A. McKenzie (MRC, Cambridge, UK). All strains are on the C57BL/6 background and Cre-negative littermate controls were used as controls. All mice were kept in a specific-pathogen-free condition and experiments were performed with 6–10-week-old female age-matched mice. Experiments were performed in accordance with federal and cantonal guidelines (Tierschutzgesetz) under permission numbers SG08/17, SG01/18, SG07/19, SG04/20, and SG01/20 following review and approval by the respective Cantonal Veterinary Offices (St. Gallen, Switzerland).

**Cells**. B16F10 cells were obtained from ATCC and cultured in complete Dulbecco's Modified Eagle's Medium (DMEM, Gibco) supplemented with 10% FCS (Sigma-Aldrich, St. Louis, Mo), 10 mmol/L NEAAs (Gibco), 1 mmol/L sodium pyruvate, 100 IU/ml penicillin/ streptomycin (Lonza, Basel, Switzerland). BHK-21 and HEK293 cells are stable transfectants carrying plasmid M369 that expresses the LCMV-glycoprotein (LCMV-GP) cDNA[58]. All cell lines were kept at 37 °C and 5% $CO_2$ in a humidified incubator and regularly examined for mycoplasma.

**Generation of B16F10-Dct$^{-/-}$ and B16F10-mCherry cells**. TRP2-deficient B16F10 cells (B16F10-*Dct*$^{-/-}$) were generated using CRISPR-Cas9 technology. Sequences for the guide RNA (sgRNA exon 2, 5′GCTTCTTCCGAT TACAGTCGGGG, Chr14; 118043326) were designed using the CHOPCHOP tool (available at http://chopchop.cbu.uib.no/index.php)[59,60]. SgRNA were inserted into the pSpCas9(BB)−2A-GFP (PX458) plasmid (Addgene, Plasmid #48138) expressing Cas9 and EGFP. B16F10 cells were transfected with the sgRNA PX458 plasmid targeting *Dct* (TRP2 gene). The *Dct* gene from cells expressing GFP was amplified (forward 5′-GCCCCTTTTTAAATCAGGAAA, Chr14:118043303-118043325 and reverse 5′-GTGATCACGTAGTCTGGATGGA, Chr14:118043127-118043149) and a T7 endonuclease I (T7EI) assay was performed to, check gene-targeting efficiency. After subcloning, single clones were again tested for deletion of *Dct* using the T7EI mismatch detection assay. Sequencing and an intracellular flow cytometric staining for the expressing of TRP2 protein (anti-TRP2/DCT ab74073, abcam, and Alexa Fluor 488–conjugated anti-rabbit IgG, Jackson Immunotools) confirmed the deletion of the *Dct* gene (Supplementary Fig. 1b). B16F10-mCherry cells were generated by transduction of B16F10 cells with lentiviral vectors carrying LeGO-C2 (Addgene, Plasmid #27339) expressing mCherry[61].

**Generation of recombinant artLCMV vectors**. Propagation-attenuated recombinant LCMV-based vectors with an artificial genome organization (artLCMV) were generated as described[30,31]. We inserted the full-length cDNA sequence of the murine melanoma-associated antigen TRP2 or of the GFP into the small segment of the LCMV genome. artLCMV vectors were generated by transient transfection of BHK21 cells stably expressing the LCMV-GP as described[58]. Virus-containing supernatant was harvested and FCS-free vector stock material was generated by infecting a suspension of HEK293 cells with a MOI of 0.001 and incubation for 72 h. Nascent viruses were harvested after centrifugation of the cells and the infectious titer was determined by focus forming assay[30,31].

**Tumor graft models and treatment with artLCMV vectors**. Mice were inoculated with $2 \times 10^5$ B16F10 melanoma cells s.c. into the flank or i.v. On day 7, when

tumors became palpable (20–50 mm$^3$), mice were treated either i.v. or i.t. with $1 \times 10^5$ FFU artLCMV-TRP2 or artLCMV-GFP. Tumor size was measured with calipers 2–3 times per week and tumor volume was calculated as width$^2 \times$ length $\times 0.5$[62,63]. Mice were sacrificed when tumor size exceeded 1000 mm$^3$ or when mice showed signs of illness. For re-challenge experiments, mice were injected on day 100 s.c. with $2 \times 10^5$ B16F10 melanoma cells into the opposite flank. For in vivo ablation of CD8$^+$ and CD4$^+$ T cells, mice were injected i.p. twice per week with 250 µg of rat anti-CD8 (clone YTS169) or rat anti-CD4 (clone YTS191) antibody starting 1 day prior to treatment with artLCMV-TRP2 and every third day afterwards. Depletion of T cell subsets was monitored in blood. For adoptive T cell transfer experiments, lymphocytes were isolated from spleens of P14 TCR tg CD45.1 mice and tumor-bearing mouse received $1–2 \times 10^5$ P14 CD8$^+$ CD45.1$^+$ T cells i.v. In addition, mice were injected i.t. with $1 \times 10^5$ FFU artLCMV-TRP2.

**Preparation of stromal cells and hematopoietic cells.** For stromal cell isolation, tumors and TDLNs were mechanically dissociated and collected in RPMI 1640 medium containing 2% FCS, 20 mM HEPES (Lonza), 1 mg/ml Collagenase P (Roche), 25 µg/ml DNaseI (Applichem) and 0.8 U/ml Dispase I (Roche). Dissociated tissue was incubated at 37 °C for 45 min, with resuspension and collection of the supernatant every 15 min. Cell suspension was filtered and washed in PBS containing 0.5% FCS and 10 mM EDTA. Stromal cells were enriched using CD45 and TER119 MACS microbeads (Miltenyi, Germany) to deplete hematopoietic cells and erythrocytes.

For the isolation of tumor-infiltrating immune cells, tumors were mechanically dissociated and disrupted on a 70 µm cell strainer (Falcon, Corning). Cells were washed once and immune cells were collected at the interface using a Ficoll gradient (Lympholyte M, CedarLane). Lymph nodes and spleens were gently smashed on a 70 µm cell strainer. Cell suspensions were directly used for flow cytometric analysis or cell sorting.

**Ex vivo restimulation and cytokine production.** For the assessment of ex vivo cytokine secretion, mice were injected i.p. with 250 µg Brefeldin A (Sigma-Aldrich) in and euthanized 4 h later[64]. Subsequent processing of tissues and staining was performed in the presence of 10 µg/ml Brefeldin A. Lymphocytes were resuspended in RMPI 1640 containing 5% FCS and stimulated with $10^{-6}$ M of the TRP2$_{180-188}$ (SVYDFFVWL) H2-K$^b$ and LCMV-NP$_{396-404}$ (FQPQNGGQFI) or LCMV-GP$_{33-41}$ (KAVYNFATC) H2-D$^b$ peptides or with 2 µg/ml TRP2 15-mer over-lapping peptide pools in the presence of 10 µg/ml Brefeldin A for 5 h at 37 °C.

**Human participants and preparation of human cells.** Patient samples were obtained from the Department of Plastic Reconstructive Surgery and the Department of Dermatology at the Kantonsspital St. Gallen. The study has been approved by the ethics committee of Eastern Switzerland (EKOS16-079, BASEC Nr. 2016-00998) and all patients provided a written informed consent. Skin specimens were taken from healthy patients undergoing abdominoplasty (female and male, age 18–61). Melanoma tissue was obtained from punch biopsies from a primary tumor of a melanoma patient. Skin and melanoma fibroblasts were isolated using the MACS Whole Skin Digestion kit (Miltenyi, Germany). Fibroblasts were cultured in RPMI 1640 with 5% FCS, 1 mmol/L sodium pyruvate, and 16 µg/ml gentamycin. Patient-matched peripheral blood mononuclear cells (PBMCs) were isolated using a Ficoll-Paque gradient centrifugation and cells were cryopreserved in 90% FCS and 10% DMSO. PBMCs were seeded in a 24-well plate at a density of $1 \times 10^6$ cells/well and fibroblasts at a density of $1 \times 10^5$ cells/well. On the following day, cells were infected with artLCMV at an MOI of 1 and incubated for 48 h. Cells were fixed for immunohistochemistry or detached for flow cytometric analysis.

**Flow cytometry.** Single cells suspensions were incubated in PBS containing 0.5% FCS and 10 mM EDTA with the indicated antibodies (Supplementary Table 1) for 20 min at 4 °C. For the identification of tetramer-binding CD8$^+$ T cells, cells were stained with PE-labeled H-2K$^b$-TRP2$_{180-188}$ tetramer (MBL, Woburn, MA) for 10 min at 37 °C. For discrimination between live and dead cells, single-cell suspensions were stained with AQUA-BV510 or e780 in PBS prior to antibody staining (Molecular Probes) or by adding 7AAD prior to acquisition (Calbiochem). For intracellular or intranuclear staining, cells were fixed and permeabilized using the BD Cytofix/Cytoperm (BD Biosciences) or the FoxP3 Transcription Factor (eBiosciences) kits, respectively. The anti-LCMV-NP antibody was conjugated with AF657 or AF594 (Antibody Labeling Kit, ThermoFisher). Cells were acquired with a BD LSRFortessa (BD Biosciences) and data analysis was performed using FACS Diva (BD Bioscience, v8.0.1) and FlowJo (Treestar Inc., v10). Cells were sorted on a Biorad S3 (Biorad) or BD Melody Cell Sorter and data were analyzed using FACSChorus (BD Biosciences, v1.3).

**Immunohistochemistry.** Tumors were harvested and fixed overnight at 4 °C in freshly prepared 4% paraformaldehyde (Merck Milipore) under agitation. Tumors were put overnight in 30% and 15% sucrose in 1xPBS, respectively and frozen in OCT. Serial sections were cut at a thickness of 8–12 µm using a cryotome. Cell layers of artLCMV-infected or uninfected human skin fibroblasts were fixed for 10 min with 4% PFA. Sections were blocked in PBS containing 10% FCS, 1 mg/ml

anti-Fcγ receptor (BD Biosciences) and 0.1% Triton X-100 (Sigma). Sections were incubated with the following primary antibodies for 1–2 h at room temperature: anti-PDPN or anti-PDPN-bio (Clone eBio8.1.1, BioLegend, 1:200), anti-GFP (Aves, 1:1000), anti-mIL-33 (R&D systems, 1:500), AF647-conjugated anti-LCMV-NP (Clone BE0106, BioXcell, 1:200), Cy3-conjugated anti-αSMA (Clone 1A4, Sigma, 1:400), and FITC-conjugated CD90 (Clone 5E10, BioLegend, 1:100). Unconjugated antibodies were detected with the following secondary antibodies: AF488-conjugated anti-chicken-IgG (Thermofisher, 1:1000), Dylight549-conjugated anti-syrian hamster-IgG, AF647-conjugated anti-goat IgG, AF488-conjugated Streptavidin (all from Jackson Immunotools, 1:500 to 1:1000). Analysis was performed using a confocal microscope (LSM-710, Carl Zeiss) and the ZEN 2010 software (Carl Zeiss, v14.0.18.201). For image analysis Imaris version 9 (Bitplane) was used.

**Single-cell RNA-seq analysis.** cDNA libraries from sorted EYFP$^+$ or PDPN$^+$ FSCs were generated on a 10x Chromium Controller (10x Genomics)[65] according to the Chromium Single-Cell 3′ Reagent Kit (v2 Chemistry) in three samples (one PBS-treated and two artLCMV-treated samples). Sequencing was run on a NovaSeq 6000 sequencing system from Illumina at the Functional Genomic Center Zurich. Initial processing of sequencing files was done running CellRanger (v2.1.1)[66] count with the Ensembl GRCm38.94 release as a reference and followed by a further quality control run in R v3.6.1 using functions from the R/Bioconductor packages scater (v1.13.15)[67] and SingleCellExperiment (v1.7.4)[68]. This quality control included the removal of damaged and contaminating cells based on (1) very high or low UMI counts (>2.5 median absolute deviation from the median across all cells), (2) very high or low total number of detected genes (>2.5 median absolute deviation from the median across all cells), (3) high mitochondrial gene content (>2.5 median absolute deviations above the median across all cells), (4) expression of one of the contamination markers *Pecam1*, *Pmel*, *Cd52*, or *Tyrp1*, (5) high expression of any of the genes *Hba-a1*, *Hba-a2*, *Krt18*, or *Lyve1*. Finally, only cells expressing mRNA encoding for the EYFP gene were retained for further analysis resulting in a dataset of 1293 cells from PBS control samples and 786 cells from artLCMV-treated samples. Downstream analysis was performed using functions from the Seurat package (SeuratWrappers, v3.1.0)[69] and included normalization, scaling, dimensional reduction with PCA and UMAP and graph-based clustering. Cluster characterization was done based on the expression of canonical marker genes as well as cluster markers calculated with the FindMarkers function of the Seurat package. In order to compare expression profiles between artLCMV-treated and PBS control samples differentially expressed genes were inferred from Wilcoxon test, again using the FindMarkers function of the Seurat package. For functional interpretation top differentially expressed genes were summarized to functionally gene signatures based on their reported gene functions. Next, cells were projected on a diffusion map[70] by utilizing the runDiffusionMap function from the scater R/Bioconductor package (v1.16.2)[67] and functional gene signatures were visualized as average expression across all genes of a signature.

**RNA isolation and quantitative real-time PCR analysis.** RNA from sorted cells was isolated using the RNeasy Mini Kit (Qiagen). Total mRNA was extracted from organs by TRIzol (Ambion) using the Direct-zol RNA Kit (ZYMO Research, USA). Contaminating DNA was eliminated through on-column DNase digestion (Zymo Research) and cDNA was generated using Quantitect Reverse Transcription Kit (Qiagen, USA). RT-PCR amplification of LCMV-NP (forward 5′ACTGACG-GAGGTCAACCCGG, reverse 5′CAAGTACTCACACGGCATGGA), *Ccl19* (QT02532173), *Cxcl13* (QT00107919), *Il33* (QT00135170), and the housekeeping genes *Hprt* (QT00166768) and TATA-binding protein (*Tbp*, QT00198443, all Qiagen) was performed in duplicates using the SYBR Green Master Mix (Applied Biosystems, Darmstadt, Germany) on a QuantStudio5 machine (Thermo Fisher, Analysis Software v1.5.1). Relative gene expression analysis was calculated using the ΔΔCt method[71]. The threshold cycle for LCMV-NP >30 cycles was determined as limit of detection.

**Statistical analysis.** Statistical analysis was performed using Prism8 (GraphPad). Unless specified otherwise, graphs depict mean ± s.d. or s.e.m. Differences between two groups were evaluated using unpaired two-tailed Student's $t$-tests. Single values of multiple groups were compared by one- or two-way analysis of variance (ANOVA), followed by Tukey's multiple comparison test or Bonferroni post hoc test. Kaplan–Meier Survival curves were assessed using Log-Rank Mantel–Cox test. Results were considered statistically significant when *$P < 0.05$, **$P < 0.01$, and ***$P < 0.001$.

**Reporting summary.** Further information on research design is available in the Nature Research Reporting Summary linked to this article.

## Data availability

scRNA-seq data are available in the ArrayExpress database with accession number: E-MTAB-9407. Ensembl GRCm38.94 was used as a reference to build index files for alignments in scRNA-seq analysis. The remaining data are available within the Article, Supplementary Information or Source Data file. Source data are provided with this paper.

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

## Acknowledgements

The authors want to thank Céline Engetschwiler, Sonja Caviezel-Firner, and Marie-Therese Abdou for excellent technical support. This study received financial support from the Swiss National Science Foundation (grants 166500, 159188 to B.L. and 157448 to L.F.), Swiss Cancer League (KLS-4409-02-2018 to L.F., KFS-4162-02-2017-R to P.K., KFS-4701-01-2019 to B.L.) and by Hookipa Pharma, Inc. (to L.F. and W.V.B.). The funders had no role in study design, data collection and analysis, decision to publish, or preparation of the manuscript.

## Author contributions

L.F., B.L. and S.S.R. designed the study, discussed data, and wrote the paper. S.S.R, J.C., L.O., A.D.M., U.M., and F.H. performed experiments, analyzed, and discussed data. C.P.-S, C.G.-C. and E.S. discussed data. P.K. discussed data and provided reagents. M.L. performed bioinformatics analyses and discussed data. A.B. provided reagents. S.S., F.S., W.V.B., and K.K.O. generated artLCMV vector material. M.S. and R.W. provided patient material.

## Competing interests

The authors declare the following competing interests. L.F. is a founder and shareholder of Hookipa Pharma Inc. F.S., S.S., and K.K.O. are employees and stock option holder of Hookipa Pharma, Inc. L.F., S.S.R, S.S., and K.K.O are listed as inventors of the patent entitled "Arenavirus particles to treat solid tumors" (patent number WO2018/185307 A1) describing the application of artLCMV vectors in the treatment of tumors. B.L., L.O., C.P.S., and C.G.C. are founders and shareholders of Stromal Therapeutics AG. S.S.R. is part-time employee of Stromal Therapeutics AG. The remaining authors declare no competing interests.
