## [Peer Review File · Nature Communications]

Reviewers' Comments:

Reviewer #1:

Remarks to the Author:

This is an interesting paper that demonstrated intratumoral injection of a recombinant lymphocytic choriomeningitis virus expressing the TRP2 melanocyte differentiation antigen resulted in T cell-dependent long-term control of melanomas in mice. In the murine studies LCMV was found to be predominantly infecting tumor cells and tumor-associated fibroblast without significant viral titers seen in tumor-draining lymph nodes. The authors used single-cell RNA-seq analysis to interrogate gene expression changes in the stromal fibroblast population and found that the virus could reprogram infected fibroblasts by activating a CXCL13-expressing FSC subset that showed an immunostimulatory signature and increased expression of the inflammatory cytokine IL-33. Indeed, ablation of IL-33 gene expression in CXCL13-Cre-positive FSCs resulted in decreased tumor-infiltrating T cell function and blocked the tumor growth inhibition seen in normal mice. The findings are important as they highlight a potential mechanism by which viral therapy can mediate efficient anti-tumor immunity and focus on stromal-derived fibroblasts, cells that have a contradictory role in tumor immunity as a rather novel part of how viral vectors may mediate tumor immunity and explains why there may be some discrepancy in the literature (e.g., since there appear to be genomically distinct FSC subpopulations). Overall, the manuscript is well written and contains an impressive amount of convincing data. My major issue relates to the generalizability of these findings and I did have a few minor comments for the authors to consider.

1. The authors suggest that their findings may apply to oncolytic viruses in general, but LCMV typically does not replicate and may be better thought of as a non-oncolytic virus. Thus, is there data to confirm that the artLCMV-based vector used in the studies does not induce oncolytic effects in melanoma cell lines in vitro? If there is no lysis, then it may not be appropriate to speculate that the mechanism identified in this report are applicable to other oncolytic viruses unless the investigators have tested other oncolytic viruses and seen a similar effect on the FSC population.
2. While it has been previously reported that LCMV can infect fibroblasts, it would be interesting for the authors to comment on how LCMV enters cells and why FSC may be more permissive for LCMV infection than other cells.
3. The investigators used an LCMV encoding the murine TRP2 antigen and show an increase in TRP-2-specific T cells, and this is solid data. They also show that the anti-tumor effect is lost in the *dct*^{-/-} mice suggesting that antigen is critical to the anti-tumor mechanism in their model. However, a control group with LCMV without TRP2 was not used; do the authors know if antigen is absolutely needed or will non-TRP-2 expressing cells induce a similar response with antigen spreading in immune competent hosts?
4. In Figure 2d the authors show LCMV in tumor cells and FSC but it is very difficult to see the LCMV staining.
5. As shown in Fig. 3 there appears to be a transition in the FSC populations in virus treated mice and the authors suggest this is a "reprogramming". However, it is not completely clear how this is happening. Is there any evidence that the virus is preferentially killing the non-CAF2 fibroblasts or is there is a true gene expression shift in a pluripotent population?

Reviewer #2:

Remarks to the Author:

This brilliant study by Ring et al showed in a very elegant way, that the fibroblastic tumor stroma is of major importance during a virus therapy in cancers. The authors found that only a local administration of their virus vector artLCMV-TRP2 was able to lead to long term control of their tumors. Mechanistically, replication of artLCMV-TRP2 in tumor stroma resulted in a specific immune signature including induction of interferons, chemokines and IL-33. By using *Cxcl13-Cre Il33 fl/fl* mice the authors found that IL-33 was one important key factor, which promoted long term control of tumors. The study is of major interest for the field, is nicely presented and well written. The authors convince with big mouse cohorts and statistical power. I however would ask to address two major points before the study is suitable for publication in nature communications.

Major comments:

1.) The authors claim that the intravenous injection is not efficient enough to reach tumor tissue and thereby fails to induce local inflammation and/or robust CD8 T cells activation. It would be very interesting if it would be possible to dissect the inflammatory component from the CD8 T cell priming. Therefore I would ask the authors how a contralateral tumor (in addition to the ipsilateral treated tumor) will behave in the case of i.T treatment. Would the contralateral tumor similarly be affected? Additionally, is the local inflammation enough to accelerate primed T cells? One could i.e. challenge mice i.t. with an artLCMV (without antigen) in addition to i.v. artLCMV-TRP2? In line with this, would a 10-fold higher intravenous dose overcome the lack of IL-33 induction and thereby be also protective. How would a 10-fold lower dose of i.T treatment look like?

2.) The authors do see also a strong anti-tumoral effect till day 20 which seems to be independent of the antigen and IL-33. Recently it was shown that Interferons under such circumstances can prevent tumor growth after arenavirus infection. Do the authors think that artLCMV-TRP2 is inducing similar effects, or is this another separate mechanism? While I do see that this topic is beyond the scope of this study the authors should at least speculate and discuss about this phenotype.

Minor comments:

There is a typo line 116 "...bolstered..."

Reviewer #3:

Remarks to the Author:

In this manuscript titled "Viral vector-mediated reprogramming of the fibroblastic tumor stroma sustains curative melanoma treatment" the authors report that intra-tumoral injection of a viral vector encoding the melanoma antigen TRP2 resulted in T cell-dependent immunity in a mouse model of injected B16F10 melanoma. Analysis of viral uptake indicated that it was predominantly uptaken by PDPN+ fibroblastic cells in the tumor microenvironment, suggesting crosstalk between CAFs and CD8+ T cells. To characterize these changes the authors performed scRNA-seq on a subset of inflammatory CAFs expressing cxcl13 and found that this population was composed of several subpopulations. They chose to further focus on CAF-derived IL-33 and showed in vivo, using transgenic mice with targeted ablation of IL-33 in cxcl13 expressing cells that the IL-33 is functionally important for sustaining T cell control of tumor growth.

The study is interesting and original, and the data is mostly of high quality. The study is of general interest in increasing our knowledge on possible routes to boost anti-tumor immunity. However, some of the conclusions are overstated, and the clinical relevance is unclear. Unfortunately, the authors disregard these limitations and do not discuss them. As such, using in the abstract and introduction phrases like: "reprogramming of FSCs by a self-antigen-expressing viral vector in the TME is critical for curative melanoma treatment.." is exaggerated, and should be toned down.

Specific comments:

1. The main concern is regarding the relevance of the findings to human disease.

All the experiments are based on i.t injections to a primary melanoma tumor, injected s.c. However, in human patients, primary melanoma tumors are surgically resected and the clinical challenge is the treatment of metastases. Will this treatment be effective/is feasible for treatment of distant metastasis (often in visceral organs, with no easy access for i.t injections? It would be important to demonstrate feasibility at least in mice, especially since the one cell line used in this study, B16F10 is metastatic.

2. The results section is written in a very succinct manner and thus some of the experiments are not well explained. For example, in line 103: "Antibody-mediated T cell depletion revealed that both CD4+ and CD8+ subsets were necessary to reject the tumor (Fig. 1e and Extended Data Fig.

1c).” What was the experiment? A few words explaining the experiments would be helpful. This is true in other experiments as well.

3. In Fig. 2h, the authors show that infection with artLCMV-TRP2 increased the fraction of PDPN+ fibroblasts, but do not suggest a mechanism. Was it due to enhanced proliferation? Other reasons? The scRNAseq data in Fig. 3 suggests that there is no significant change in CAF proliferation.

4. In line 175: “The transformation of the FSC landscape with the acquisition of immunostimulatory properties was confirmed..”. This term is used again in the discussion (line 390). The use of the term “transformation” in cancer context is usually reserved to transition to malignancy, and is therefore confusing when describing the effect of viral transduction on (non-transformed) fibroblasts.

5. How did the authors choose to focus on cxcl13? What was the rationale? Did they first analyze expression of chemokines other than cxcl13 and ccl19 in a non-biased manner? What other T cell chemoattractants were analyzed, if at all? As it is, the choice seems rather random.

6. The authors performed scRNAseq on a fraction of immunostimulatory cxcl13+ fibroblasts, and found within them subsets of myCAF-like fibroblasts (Fig. 3). Since previous literature suggests that inflammatory CAFs are distinct from myCAFs, this finding requires further discussion.

7. Similar to comment #5 above, the selection of IL-33 to focus on is also not explained. Other genes that are highlighted in Fig. 3h would be just as valid candidates to look at CAF-mediated T cells modulation (e.g cxcl9, cxcl10).

8. IL-33 is known to mediate Th2 immunity, and to activate ILC2 cells. It was recently suggested that recruitment of CD8+ T cells to PDAC is mediated via IL-33 stimulated ILC2 cells (Moral JA Nature 2020, and Cancer Discovery 2020 DOI: 10.1158/2159-8290.CD-RW2020-032). Did the author analyze whether the effect of IL-33 on CD8+ T cells is mediated via ILC2 cells? Moreover, a recent study on CAF-derived IL-33 suggested that it promoted tumor metastasis by modulating Th2 immunity (Shani et al. Cancer Res. 2020). In this context, the authors should address the modulation of the immune milieu (not limited to CD8+ T cells) in the transplanted tumors in response to IL-33 secretion, to better dissect the mechanism by which IL-33 affects CD8+T cell function and prevent their exhaustion.

9. The relevant recent IL-33 literature should be cited.

10. Cxcl13 is not fibroblast specific. It is expressed by many other cells, including T cells and dendritic cells. Therefore, crossing Cxcl13-Cre/tdTom mice to Il33fl/fl, to generate Cxcl13-Cre Il33fl/fl mice would result in IL-33 ablation in multiple cell types, and is not CAF specific. This limitation is completely ignored by the authors in their suggested mechanism. While fibroblast-specific Cre mice are problematic (and beyond the scope), this limitation and alternative interpretation of the results must be addressed.

Reviewer #4:

Remarks to the Author:

This study by Ring et al presents some interesting data identifying a critical role for fibroblast reprogramming in sustained response to anti-melanoma vaccination. These data are novel and likely to be of significant interest to the Nature Communications readership. My main concern is a lack of detail regarding why/how the viral vector used elicits the fibroblast reprogramming reaction, which may limit the impact these findings could have and their potential application.

Major points:

1. The terminology used to describe the fibroblast subpopulations identified needs to be addressed.

- The markers used for the myCAF2 designation are questionable and not consistent with the studies that originally defined the MyoCAF term. In Extended Data Figure 6 the markers shown for

the MyoCAF2 population includes multiple genes upregulated following hypoxia, including Vegfa. This is consistent with the VEGF+ CAF subpopulation described by Grauel et al. (NCOMMS 2020). The authors should amend the terminology used to reflect this.

2. How does the intratumoural artLCMV-TRP2 treatment modify myoCAF features of the tumour microenvironment?

- The UMAP plot (Figure 3a) presented suggests a phenotypic shift in the myoCAF1 population. The authors should present data on Differential expression analysis between PBS and artLCMV-TRP2 treated myoCAF1, to determine whether the changes in gene expression are similar or different between iCAFs and myoCAFs. Given that myoCAFs have been shown to be the principal CAF subpopulation involved in immune checkpoint non-response, if these cells can also be reprogrammed to an IL-33+ phenotype this would significantly enhance the potential reach of the findings presented in this paper.

3. Does Cxcl13-IL-33fl/fl impact the phenotype of stromal cells?

- In Figure 3 the authors show that artLCMV-TRP2 treatment causes fibroblasts to upregulate multiple genes that could be involved in the recruitment and activation of CD8+ T-cells. Figure 4 shows that intratumoural artLCMV-TRP2 treatment in Cxcl13-IL-33fl/fl mice fails to control tumour growth. The conclusions drawn suggest that this is due to the direct action of CD8+ T-cells by IL-33. However, an alternative explanation could be that artLCMV-TRP2 treatment induced IL-33 is responsible for the fibroblast reprogramming. Experiments should be performed to determine whether IL-33 acts directly through CD8 activation, indirectly by reprogramming fibroblasts or a combination of the two mechanisms.

4. Could intratumoural artLCMV-mediated fibroblast reprogramming also be used as an immunotherapy adjunct?

- Grauel et al. (NCOMMS 2020) showed that reprogramming the stroma using TGF-beta blockade generated a fibroblast subpopulation, similar to that described in this study following artLCMV-TRP2 treatment, which increased the efficacy of PD1 immunotherapy. Could combined artLCMV treatment and PD1 immunotherapy be an attractive strategy for patients without vaccine actionable mutations?

5. Is the fibroblast reprogramming described specific to the Trp2 antigen or artLCMV vectors?

- All the experiments analysing fibroblast phenotypes are compared to PBS controls. What happens to fibroblast phenotypes when transduced with another antigen or a different type of viral vector?

Minor Points:

There are some errors in the legend for figure 3. For example, Panel k is referred to when describing the statistical tests used but there is no panel k.

Point-by-point reply

REVIEWER COMMENTS

Reviewer #1 (Remarks to the Author): with expertise in melanoma and viral-based immunotherapy

This is an interesting paper that demonstrated intratumoral injection of a recombinant lymphocytic choriomeningitis virus expressing the TRP2 melanocyte differentiation antigen resulted in T cell-dependent long-term control of melanomas in mice. In the murine studies LCMV was found to be predominantly infecting tumor cells and tumor-associated fibroblast without significant viral titers seen in tumor-draining lymph nodes. The authors used single-cell RNA-seq analysis to interrogate gene expression changes in the stromal fibroblast population and found that the virus could reprogram infected fibroblasts by activating a CXCL13-expressing FSC subset that showed an immunostimulatory signature and increased expression of the inflammatory cytokine IL-33. Indeed, ablation of IL-33 gene expression in CXCL13-Cre-positive FSCs resulted in decreased tumor-infiltrating T cell function and blocked the tumor growth inhibition seen in normal mice. The findings are important as they highlight a potential mechanism by which viral therapy can mediate efficient anti-tumor immunity and focus on stromal-derived fibroblasts, cells that have a contradictory role in tumor immunity as a rather novel part of how viral vectors may mediate tumor immunity and explains why there may be some discrepancy in the literature (e.g., since there appear to be genomically distinct FSC subpopulations). Overall, the manuscript is well written and contains an impressive amount of convincing data. My major issue relates to the generalizability of these findings and I did have a few minor comments for the authors to consider.

1. The authors suggest that their findings may apply to oncolytic viruses in general, but LCMV typically does not replicate and may be better thought of as a non-oncolytic virus. Thus, is there data to confirm that the artLCMV-based vector used in the studies does not induce oncolytic effects in melanoma cell lines in vitro? If there is no lysis, then it may not be appropriate to speculate that the mechanism identified in this report are applicable to other oncolytic viruses unless the investigators have tested other oncolytic viruses and seen a similar effect on the FSC population.

We are glad that R1 appreciates that the work highlights a potential mechanism by which virus-based vectors mediate efficient antitumor immunity. Indeed, the main message conveyed in our manuscript is that the local reprogramming of a specific FSC subset by a non-cytopathic viral vector fuels the functional activation of antitumor T cells and thereby facilitates curative treatment.

The vaccine vector used in this study is based on the non-cytopathic lymphocytic choriomeningitis virus (LCMV) that can propagate in target cells without causing direct cytopathic effects. Wildtype LCMV can efficiently infect different types of cancer cells and replicate without causing a cytopathic effect (Kalkavan et al. 2018 PMID 28248314). The LCMV-based vectors are propagation-attenuated and exhibit a significantly reduced propagation capacity (Kallert et al. 2017 PMID 28548102). To demonstrate that the LCMV-based vectors do not influence the viability of infected fibroblasts, we obtained CD45⁻ CD31⁻ EpCAM⁻ Ter119⁻ cells from mouse back skin and incubated them with artLCMV-GFP at an MOI of 0.1 for 48 hrs. Skin-derived fibroblasts were characterized by the expression of the FSC markers PDPN and CD90 (Fig. R1.1a and b for the attention of the reviewer). As expected, infection with artLCMV vectors neither altered the morphology of the infected fibroblasts (Fig. R1.1a) nor resulted in a higher frequency of apoptotic cells (Fig. R1.1c). These observations are further supported by our data of human skin-derived fibroblasts shown in Fig. 3e-g.

Figure R1.1. LCMV-based vectors are non-cytopathic and do not influence the viability of in vitro cultured skin fibroblasts. (a) Morphology of cultured mouse skin $CD31^- CD45^- EpCAM^- Ter119^-$ cells incubated for 48 hrs with artLCMV-GFP at an MOI of 0.1. Scale bar 20 μ m. (b) Representative plot (c) Frequency of viable and apoptotic cells in $CD90^+ PDPN^+$ fibroblasts.

2. While it has been previously reported that LCMV can infect fibroblasts, it would be interesting for the authors to comment on how LCMV enters cells and why FSC may be more permissive for LCMV infection than other cells.

We agree with the reviewer that different studies have convincingly shown that fibroblastic reticular cells in secondary lymphoid organs are a significant target of LCMV (Mueller et al. 2007 PMID 17878315, Bonilla et al. 2012 PMID 22323740, Perez-Shibayama et al. 2020 PMID 32917792) or LCMV-based vectors (Kallert et al. 2017 PMID 28548102). Fibroblasts in many tissues express α -dystroglycan, the receptor for LCMV host cell-entry (Durbeej et al., 1998 PMID 9524190). The data shown in Figure 3 of the manuscript indicate that LCMV-based vectors also exhibit a high tropism for tumor-associated fibroblasts. Transcriptomic analysis of EYFP⁺ FSCs obtained from Cxcl13-Cre EYFP mice revealed that α -dystroglycan (encoded by *Dag1*) is expressed at high levels on tumor-associated FSCs (see R1.2). LCMV infects a broad range of host cells including myeloid cells, endothelial cells, fibroblasts and tumor cells. In contrast to myeloid cells, fibroblasts are long-lived cells and thus the artLCMV-delivered antigens persist within the TME until $CD8^+$ T cells clear infected cells within the tumor tissue. Moreover, confocal microscopic analysis unveiled that LCMV-transduced PDPN⁺ FSCs were localized mainly in the tumor margin, a site where they can efficiently orchestrate interactions between infiltrating immune cells and tumor cells.

Figure R1.2. The receptor for LCMV host cell-entry is expressed on tumor-associated FSCs. UMAP feature plots of *Dag1* expression in EYFP⁺ FSCs.

3. The investigators used an LCMV encoding the murine TRP2 antigen and show an increase in TRP2-specific T cells, and this is solid data. They also show that the anti-tumor effect is lost in the *dct*^{-/-} mice suggesting that antigen is critical to the anti-tumor mechanism in their model. However, a control group with LCMV without TRP2 was not used; do the authors know if antigen is absolutely needed or will non-TRP-2 expressing cells induce a similar response with antigen spreading in immune competent hosts?

We are glad that R1 acknowledges the causal connection between TRP2-specific T cells and tumor rejection following i.t. treatment of artLCMV-TRP2. To demonstrate that the vector-delivered self-antigen TRP2 is crucial, we have generated a B16F10 cell line that lacks the TRP2 antigen (B16F10-*Dct*^{-/-}, please see ED Fig. 1b) and found that TRP2 antigen expression by the tumor cells is crucial for T cell-dependent tumor rejection (Fig. 1d). Moreover, we treated mice with LCMV-based vectors expressing an irrelevant antigen (artLCMV-GFP) and found that tumor growth is initially delayed, but tumors are not rejected (Fig. 1e). Thus, expression of the TRP2 melanoma antigen by the tumor cells as well as by the vaccine vector is critical to achieve complete tumor rejection following i.t. artLCMV-TRP2 treatment. Please see also our response to Point 4 of R4.

4. In Figure 2d the authors show LCMV in tumor cells and FSC but it is very difficult to see the LCMV staining.

LCMV-NP⁺ cells are displayed in white and highlighted by arrows/ arrowheads (Fig. 3d). We show here the staining control from a PBS-treated control mice to better distinguish LCMV-NP⁺ cells (Fig. R1.3 for the attention of the reviewer).

Figure R1.3. LCMV-NP staining in artLCMV-TRP2 treated tumor. Confocal microscopy of tumors. Representative area in the tumor centre from artLCMV-TRP2 or PBS-injected mice (see Fig. 3d in the manuscript).

5. As shown in Fig. 3 there appears to be a transition in the FSC populations in virus treated mice and the authors suggest this is a “reprogramming”. However, it is not completely clear how this is happening. Is there any evidence that the virus is preferentially killing the non-CAF2 fibroblasts or is there is a true gene expression shift in a pluripotent population?

Non-cytopathic LCMV does not directly lyse target cells and infected cells are cleared by specific T cells (see Point 1 R1). Thus, we exclude the possibility that artLCMV vectors are directly killing any CAF population in the TME. We have selected an early time point (day 11; day 4 after i.t. artLCMV-TRP2 treatment) for the transcriptomic analysis of the tumor FSCs to specifically study artLCMV-induced changes in the TME that lead to the formation of immune cell nurturing niches and foster the

maintenance of antitumoral immunity.

While subset identity was maintained in each FSC cluster, the artLCMV-induced shift from iCAF1 to iCAF2 reveals a pronounced cellular activation program of the iCAF immune-stimulatory subset indicating a true gene expression shift. The distinct immune-stimulatory state in the iCAF2 cluster is characterized by highly elevated expression of chemokines and cytokines, antigen presentation and IFN type I signature. Our data corroborate the plasticity described for fibroblasts in lymphoid organs, peripheral tissues or tumors described by other recent studies (Perez-Shibayama et al. 2020 PMID 32917792, Cupovic et al., accompanying manuscript, Grauel et al. 2020 PMID 33298926).

We conclude that the immune-stimulatory iCAFs foster antitumoral immunity at different levels: recruitment and proper positioning of immune cells within the TME, preservation of T cell functionality in specific niches in the TME, increased antigen presentation activity and transient persistence of the viral vectors and the delivered tumor antigen in the tumor tissue. Our data support the notion that the viral vector-induced shift of the iCAF subset towards a strong immune-stimulatory state fosters the functionality of antitumoral T cells and prevents their exhaustion.

Reviewer #2 (Remarks to the Author): with expertise in LCMV and cancer

This brilliant study by Ring et al showed in a very elegant way, that the fibroblastic tumor stroma is of major importance during a virus therapy in cancers. The authors found that only a local administration of their virus vector artLCMV-TRP2 was able to lead to long term control of their tumors. Mechanistically, replication of artLCMV-TRP2 in tumor stroma resulted in a specific immune signature including induction of interferons, chemokines and IL-33. By using Cxcl13-Cre Il33 fl/fl mice the authors found that IL-33 was one important key factor, which promoted long term control of tumors. The study is of major interest for the field, is nicely presented and well written. The authors convince with big mouse cohorts and statistical power. I however would ask to address two major points before the study is suitable for publication in nature communications.

Major comments:

1. The authors claim that the intravenous injection is not efficient enough to reach tumor tissue and thereby fails to induce local inflammation and/or robust CD8 T cells activation. It would be very interesting if it would be possible to dissect the inflammatory component from the CD8 T cell priming. Therefore I would asked the authors how a contralateral tumor (in addition to the ipsilateral treated tumor) will behave in the case of i.T treatment. Would the contralateral tumor similarly be affected? Additionally, is the local inflammation enough to accelerate primed T cells? One could i.e. challenged mice i.t. with an artLCMV (without antigen) in addition to i.v. artLCMV-TRP2? In line with this, would a 10-fold higher intravenous dose overcome the lack of IL-33 induction and thereby be also protective. How would a 10-fold lower dose of i.T treatment look like?

We agree with the reviewer that the elaboration of the spatiotemporal inflammatory component after i.t. artLCMV-TRP2 treatment is an important question and we appreciate the experimental suggestions. First, we analysed the antitumor effect in a bilateral s.c. B16F10 tumor model as outlined in Fig. R2.1a. Interestingly, we observed a substantially delayed growth of the non-injected contralateral tumors in mice injected i.t. with artLCMV-TRP2 compared to PBS-treated mice (Fig. 2.1a to c). Moreover, pulmonary metastasis were significantly reduced in mice following i.t. artLCMV-TRP2 treatment in the accessible s.c. primary tumor (see new Fig. 1h and i and Point 1 R3). These additional data further support our main conclusion that tumor-associated fibroblasts can improve the functional activation and maintenance of tumor-infiltrating T cells. Thus, the artLCMV-induced antitumor immunity is not confined to the primary tumor, but T cells imprinted by a reprogramed TME also mediate antitumor immunity at secondary distant tumor sites indicating the translational relevance to treat cancer patients with distinct metastatic lesions.

Figure R2.1. Intratumoral artLCMV-TRP2 injection inhibits tumor growth in secondary distant tumors. (a) Treatment scheme for a bilateral B16F10 tumor challenge. (b-c) Tumor kinetics of (b) the ipsilateral tumor injected i.t. with artLCMV-TRP2 and (c) the contralateral non-injected tumor. Pooled data from two independent experiments with $n=12$ (PBS) and $n=13$ (artLCMV-TRP2, i.t.) mice.

To investigate whether the local inflammation is sufficient to foster TRP2-specific T cells induced by the intravenous route, we treated mice i.t. with artLCMV-GFP and i.v. with artLCMV-TRP2. Tumor growth was significantly delayed in the artLCMV-GFP, i.t. / artLCMV-TRP2, i.v.-treated mice compared to mice injected either with artLCMV-GFP, i.t. or artLCMV-TRP2, i.t. suggesting that reprogramming of the local environment fosters infiltrating T cells primed via the i.v. route (Fig. R2.2a). However, tumor rejection and survival of artLCMV-GFP, i.t. / artLCMV-TRP2, i.v.-treated mice was reduced compared to i.t. artLCMV-TRP2 therapy (Fig. R2.2b).

Lastly, we performed an experiment using different doses of artLCMV-TRP2 injected either via the i.t. or the i.v. route and found that it is rather the route of injection than the amount of virus-based particles that determines antitumor immunity (Fig. R2.2c). We consider this is an important message to convey and have therefore incorporated this data set into ED Figure 1a. Overall, these novel data confirm that reprogrammed Cxcl13-Cre⁺ FSCs form niches in the TME that support accumulation and sustenance of antitumoral T cells following i.t. but not i.v. artLCMV-TRP2 treatment.

Figure R2.2. Intratumoral artLCMV-TRP2 injection is superior to i.v. artLCMV-TRP2 in mediating tumor rejection. (a and b) Mice were injected s.c. with 2×10^5 B16F10 and treated either i.t. or i.v. with artLCMV-GFP or artLCMV-TRP2. (a) Tumor kinetic and (b) survival of mice. (c) Mice were injected with 2×10^5 B16F10 and treated either i.t. or i.v. with different doses of artLCMV-TRP2.

2. The authors do see also a strong anti-tumoral effect till day 20 which seems to be independent of the antigen and IL-33. Recently it was show that Interferons under such circumstances can prevent tumor growth after arenavirus infection. Do the authors think that artLCMV-TRP2 is inducing similar effects, or is this another separate mechanism? While I do see that this topic is beyond the scope of this study the authors should at least speculate and discuss about this phenotype.

We agree with the reviewer that additional immune mechanisms such as type I interferons are important to control tumor growth after artLCMV treatment but complete rejection of tumors after i.t. treatment with artLCMV-TRP2 was fully dependent on self-antigen expression by the tumor cells (see Fig. 1d) and self-antigen-delivery by the vector (Fig. 1e). Please see also our response to Point 3 of R1.

However, the reviewer raises a valid point considering the important role of type I IFNs in LCMV-induced antitumor immunity, which we had already considered and had performed experiments that were not included in the manuscript. We found that impaired IFNAR signaling in B16F10 cells (B16F10-*IFNAR1*^{-/-}) did not influence tumor growth delay (Fig. R2.3a), whereas *IFNAR1*-deficiency in host cells significantly inhibited antitumor immunity following i.t. artLCMV-TRP2 treatment (Fig. R2.3b). Moreover, we found that IFNAR-signaling on both compartments, the hematopoietic and the non-hematopoietic compartment, was crucial for tumor rejection after i.t. artLCMV-TRP2 treatment (Fig. R2.3c).

A recent study by our group revealed that type I IFN stimulation of lymph node FRCs during acute LCMV infection is critical to prevent exhaustive CD8⁺ effector T cell activation and thereby maintains antiviral CD8⁺ effector T cell differentiation (Perez-Shibayama et al. 2020 PMID 32917792). Likewise, the transcriptomic analysis of tumor-associated FSCs revealed that artLCMV-induced immunostimulatory gene signatures including type I IFN and interferon-stimulated genes was pronounced in the iCAF cluster (Fig. 5e and f, ED Fig. 6c). Despite the interesting role of type I IFN and IFN I-induced molecules such as the CXCR3 ligands CXCL9 and CXCL10 in antitumor immunity, we have decided to focus our analysis on the function of the stromal-derived pro-inflammatory molecule IL-33 to avoid excessive display of data in the manuscript.

In sum, our data unveil that artLCMV poise an immune-stimulatory state in iCAFs and thereby foster immune cell composition as well as functional activation of T cells, which is an important prerequisite to steer efficient and persistent antitumoral immunity.

Figure R2.3. Type I IFNAR1 signalling in host cells is crucial in antitumor immunity following i.t. artLCMV-TRP2. (a) Mice were injected s.c with 2×10^5 B16F10 or B16F10-IFNAR1^{-/-} cells and treated i.t. with artLCMV-TRP2 on day 7. (b) WT or IFNAR1-deficient mice were injected s.c with 2×10^5 B16F10 and treated i.t. with artLCMV-TRP2 on day 7. (c) Bone marrow chimeric mice were generated by transferring IFNAR1-proficient (WT) or IFNAR1-deficient bone marrow into sub-lethally irradiated WT or IFNAR1^{-/-} mice that were inoculated s.c. with 2×10^5 B16F10. Mice were treated i.t. with artLCMV-TRP2 and tumor growth was monitored. Pooled data from two independent experiments with n=12 to 14 mice (a) and n=6 to 15 mice (b) or three independent experiments with n=12 to 19 mice (c).

Minor comments: There is a typo line 116 "...bolstered..."

We thank the reviewer for careful reading of the manuscript and have corrected this typo.

Reviewer #3 (Remarks to the Author): with expertise in cancer associated fibroblasts

In this manuscript titled ““Viral vector-mediated reprogramming of the fibroblastic tumor stroma sustains curative melanoma treatment” the authors report that intra-tumoral injection of a viral vector encoding the melanoma antigen TRP2 resulted in T cell-dependent immunity in a mouse model of injected B16F10 melanoma. Analysis of viral uptake indicated that it was predominantly taken up by PDPN+ fibroblastic cells in the tumor microenvironment, suggesting crosstalk between CAFs and CD8+ T cells. To characterize these changes the authors performed scRNA-seq on a subset of inflammatory CAFs expressing cxcl13 and found that this population was composed of several subpopulations. They chose to further focus on CAF-derived IL-33 and showed in vivo, using transgenic mice with targeted ablation of IL-33 in cxcl13 expressing cells that the IL-33 is functionally important for sustaining T cell control of tumor growth.

The study is interesting and original, and the data is mostly of high quality. The study is of general interest in increasing our knowledge on possible routes to boost anti-tumor immunity. However, some of the conclusions are overstated, and the clinical relevance is unclear. Unfortunately, the authors disregard these limitations and do not discuss them. As such, using in the abstract and introduction phrases like: “reprogramming of FSCs by a self-antigen-expressing viral vector in the TME is critical for curative melanoma treatment..” is exaggerated, and should be toned down.

Specific comments:

1. The main concern is regarding the relevance of the findings to human disease. All the experiments are based on i.t injections to a primary melanoma tumor, injected s.c. However, in human patients, primary melanoma tumors are surgically resected and the clinical challenge is the treatment of metastases. Will this treatment be effective/is feasible for treatment of distant metastasis (often in visceral organs, with no easy access for i.t injections? It would be important to demonstrate feasibility at least in mice, especially since the one cell line used in this study, B16F10 is metastatic.

We agree with the reviewer that patients usually exhibit distant metastasis often not accessible for i.t. injections. To address the clinical relevance of local viral vector administration, we assessed the systemic antitumor response in distant pulmonary metastasis in mice treated i.t. with artLCMV-TRP2 into the s.c. tumor (Fig. R3.1a). The novel data unveil that immune responses elicited by local artLCMV-TRP2 treatment are not limited to the injected s.c. tumor but significantly reduced tumor growth in the lung (Fig. R3.1b). Thus, these data provide evidence that reprogramming of the local tumor-associated fibroblastic stroma by artLCMV bolsters effector functions of tumor-specific T cells, which can traffic to distant metastatic lesions to exert antitumor activity. We consider this important for the translational relevance and have added this data set in the revised Fig. 1h and i in the manuscript. Please note that we show in the revised Fig. 1g that all mice cured from primary melanomas after i.t. artLCMV-TRP2-treatment were completely protected from a re-challenge with B16F10 into the opposite flank demonstrating the generation of a systemic immunological memory response.

Moreover, we would also bring to the attention our response to Point 1 of Reviewer 2 where we observed significantly delayed tumor growth of a contralateral implanted tumor in mice injected i.t. with artLCMV-TRP2 into the ipsilateral tumor (Fig. R2.1).

Figure R3.1. Intratumoral artLCMV-TRP2 injection inhibits metastatic tumors growing distant non-accessible organs. (a) Treatment scheme for a metastatic B16F10 tumor challenge. (b and c) Representative pictures and (c) enumeration of pulmonary nodules. Pooled data from three independent experiments with $n=10$ (PBS) and $n=11$ (artLCMV-TRP2, i.t.) mice.

2. The results section is written in a very succinct manner and thus some of the experiments are not well explained. For example, in line 103: “Antibody-mediated T cell depletion revealed that both $CD4^+$ and $CD8^+$ subsets were necessary to reject the tumor (Fig. 1e and Extended Data Fig. 1c).” What was the experiment? A few words explaining the experiments would be helpful. This is true in other experiments as well.

We have adjusted our wording and describe more accurately the experimental setting.

3. In Fig. 2h, the authors show that infection with artLCMV-TRP2 increased the fraction of PDPN+ fibroblasts, but do not suggest a mechanism. Was it due to enhanced proliferation? Other reasons? The scRNAseq data in Fig. 3 suggests that there is no significant change in CAF proliferation.

PDPN can be upregulated in disease and a multitude of stimuli can promote enhanced PDPN expression including pro-inflammatory cytokines and pro-tumorigenic signals (Astarita et al. 2012 PMID 22988448) and infection of mouse skin fibroblasts with artLCMV-GFP resulted in increased PDPN expression (Fig. R3.2a). Indeed, we did not observe differences in the abundance of proliferating CAFs between PBS- and artLCMV-treated mice. However, we found increased expression levels of PDPN in iCAFs (ED Fig. 5c). Moreover, we found significantly higher expression of PDPN on tumor-associated FSCs in artLCMV-treated mice supporting our finding that LCMV vector administration induces expansion of iCAFs in the TME (Fig. R3.2b and c).

Figure R3.2. PDPN is upregulated in fibroblasts upon artLCMV infection in vivo and in vivo. (a) Mouse skin fibroblasts were incubated for 48 hrs with artLCMV-GFP at an MOI of 0.1 and mean expression of PDPN within $CD90^+$ $PDPN^+$ cells was measured. (b and c) Mean expression of PDPN on tumor-associated $PDPN^+$ (b) and $EYFP^+$ $PDPN^+$ FSCs (c).

4. In line 175: “The transformation of the FSC landscape with the acquisition of immunostimulatory properties was confirmed..”. This term is used again in the discussion (line 390). The use of the term “transformation” in cancer context is usually reserved to transition to malignancy, and is therefore confusing when describing the effect of viral transduction on (non-transformed) fibroblasts.

We have adjusted our wording and now use the words remodeling or reprogramming.

5. How did the authors choose to focus on *cxcl13*? What was the rationale? Did they first analyze expression of chemokines other than *cxcl13* and *ccl19* in a non-biased manner? What other T cell chemoattractants were analyzed, if at all? As it is, the choice seems rather random.

Secondary lymphoid organs (SLO) as well as peripheral organs are underpinned by highly specialized fibroblastic stromal cells that form dedicated microenvironmental niches to govern, regulate and sustain innate and adaptive immunity. Immune-interacting FSCs are characterized by the lymphocyte-attracting chemokines *Ccl19* and *Cxcl13* (Chai et al. 2013 PMID 23623380, Onder et al. 2018 PMID 28709801, Pikor et al. 2020 PMID 32424359) and play crucial roles in governing immune responses in infections and cancer (Cupovic et al. 2016 PMID 26921107, Gil-Cruz et al. 2016 PMID 27798617, Perez-Shibayama et al. 2018 PMID 30097537, Cheng et al. 2018 PMID 29391257). Moreover, *Ccl19*- and *Cxcl13*-expressing fibroblasts underpin tertiary lymphoid structures (TLS), which are associated with improved efficacy in checkpoint inhibitor therapy in cancer patients (Cabrita et al. 2020 PMID 31942071).

The immune-interacting chemokine *Cxcl13* is expressed in skin-derived fibroblasts (see Fig. R3.3), and was recently confirmed by Buechler et al. 2021 (PMID 33981032). Moreover, we found that *Cxcl13* is expressed in tumor-associated FSCs and administration of artLCMV induced higher expression of *Cxcl13* in iCAFs (new Fig. 4b and e and Fig. 5g). Thus, we considered it reasonable to utilize the transgenic *Cxcl13*-Cre/tdTomato R26R-EYFP mouse model to study a population of immune-interacting *Cxcl13*-expressing FSCs in orthotopically implanted B16F10 melanomas. Please note that we now provide a more detailed description of the *Cxcl13*-Cre/tdTomato R26R-EYFP mouse model in the manuscript and in Point 6.

Figure R3.3. *Cxcl13* is expressed in skin fibroblasts. (a) EYFP expression in *Cxcl13*-Cre/tdTom EYFP and Cre-Ctrl mice (gated on Live Ter119⁻ cells). (b-e) Gating scheme for skin fibroblasts from *Cxcl13*-Cre/tdTom EYFP mice found within Live Ter119⁻ CD45⁻ EpCAM⁻ (b) and CD31⁻ cells (c). (d) EYFP⁺ PDPN⁺ and EYFP⁺ PDPN⁻ cells marking FSCs with current and past *Cxcl13* expression. (e) TdTom⁺ cells within EYFP⁺ PDPN⁺ cells labelling FSCs with current CXCL13 expression.

6. The authors performed scRNAseq on a fraction of immunostimulatory *cxcl13*⁺ fibroblasts, and found within them subsets of myCAF-like fibroblasts (Fig. 3). Since previous literature suggests that inflammatory CAFs are distinct from myCAFs, this finding requires further discussion.

We thank the reviewer for this comment, which is also a reflection of the current discussion about fibroblast heterogeneity in cancer and other inflammatory diseases (Sahai et al. 2020 PMID 31980749, Davidson et al. 2021 PMID 33911232). In the *Cxcl13*-Cre/tdTomato R26R-EYFP mouse model (Onder et al. 2017 PMID 28709801, Pikor et al. 2020 PMID 32424359) the Cre recombinase expression is under the control of the *Cxcl13* promoter and allows EYFP-labelling of cells with current *Cxcl13* expression, cells with past chemokine expression or the progeny of chemokine-expressing cells. Thus, EYFP⁺ cells comprise different CAF populations in the TME with current or past EYFP expression including FSCs with immune-stimulatory properties (iCAFs) but also myCAFs with traits of ECM remodeling or immune-suppressive features. Flow cytometric analysis showed that iCAFs, identified by expression of CD26, CD34, Sca-1 and Ly6C, represent 60% of the EYFP⁺ PDPN⁺ fibroblasts indicating that the *Cxcl13*-Cre targets iCAFs. In addition, expression of the reporter fluorescent protein tdTomato falls under the control of the *Cxcl13* promoter facilitating detection of current *Cxcl13* expression. Our transcriptomic and flow cytometric analysis revealed that *Cxcl13* mRNA expression and tdTom/ CXCL13 protein expression was almost exclusively restricted to iCAF2 in artLCMV-injected tumors.

We are aware of the limitations of transgenic mouse models. However, there is currently no mouse model available that allows for targeting of a specific CAF subset. Thus, we considered the *Cxcl13*-Cre/tdTom EYFP mouse model a powerful tool to study the potential of immune-stimulatory *Cxcl13*-expressing iCAFs in the TME. The *Cxcl13*-Cre/tdTom EYFP mouse model allowed us to resolve the molecular circuits underlying the artLCMV-induced activation of FSC landscape in the TME and demonstrated that iCAF-derived molecules such as IL-33 play a crucial role in the TME by sustaining T cell functionality. We have amended the manuscript and provide a more detailed description of the *Cxcl13*-Cre/tdTom EYFP mouse model to better understand the artLCMV-induced remodeling of the tumor FSC landscape.

7. Similar to comment #5 above, the selection of IL-33 to focus on is also not explained. Other genes that are highlighted in Fig. 3h would be just as valid candidates to look at CAF-mediated T cells modulation (e.g *cxcl9*, *cxcl10*).

We agree with the reviewer that there are other interesting genes upregulated in the tumor-associated iCAFs after i.t. artLCMV treatment. However, the role of some candidates is well studied in antitumor immunity such as antigen-presentation molecules, molecules in the type I IFN pathway or T cell chemoattractants such as *Cxcl9* and *Cxcl10*.

While IL-33 has long been studied in the context of Th2-induced immunopathologies, its functional role in the stimulation of Th1-related immune responses is not well established. Likewise, IL-33 exhibits a similar dichotomy in cancer immunity displaying both pro- and antitumoral functions dependent on the specific tumor microenvironment. Our data show a spatiotemporal fluctuation in the expression of ST2, the receptor for IL-33, on tumor-infiltrating CD8⁺ T cells starting at the time point when CD8⁺ T cells migrate into the tumor (d11 to d13). Moreover, we found that *Cxcl13*-Cre *Il33*^{fl/fl} mice failed to control tumor growth suggesting an immune-supporting function of IL-33. Thus, we considered it reasonable to study FSC-derived IL-33 and its functional influence on tumor-specific T cells in the context of LCMV-induced antitumor immunity. Future studies will be required to elaborate the full spectrum of mechanism underpinning iCAF-T cell interactions in the TME resulting in further interesting candidates

for tumor therapy.

8. IL-33 is known to mediate Th2 immunity, and to activate ILC2 cells. It was recently suggested that recruitment of CD8+ T cells to PDAC is mediated via IL-33 stimulated ILC2 cells (Moral JA Nature 2020, and Cancer Discovery 2020 DOI: 10.1158/2159-8290.CD-RW2020-032). Did the author analyze whether the effect of IL-33 on CD8+ T cells is mediated via ILC2 cells?

Moreover, a recent study on CAF-derived IL-33 suggested that it promoted tumor metastasis by modulating Th2 immunity (Shani et al. Cancer Res. 2020). In this context, the authors should address the modulation of the immune milieu (not limited to CD8+ T cells) in the transplanted tumors in response to IL-33 secretion, to better dissect the mechanism by which IL-33 affects CD8+T cell function and prevent their exhaustion.

To address this comment, we assessed if the effect of FSC-derived IL-33 is directly acting on CD8⁺ T cells or is mediated by an indirect ILC2/ CD8⁺ T cells cross-interaction by (i) analyzing the abundance of ILCs in the TME (Fig. R3.3), (ii) by analyzing the tumor immune milieu in Cxcl13-Cre *I133^{fl/fl}* and Cre-negative Ctrl mice (ED Fig. 8 and Fig. R3.4) and (iii) by performing adoptive T cell transfer experiments (Fig. R4.3). We examined tumor-infiltrating CD127⁺ CD25⁺ ILCs and CD127⁺ CD25⁺ ST2⁺ ILC2s in artLCMV-TRP2-treated Cxcl13-Cre *I133^{fl/fl}* and Cre-negative Ctrl on day 11 and day 15 (day 4 and day 8 after artLCMV treatment). At both time points, we found very low but comparable frequencies of ILCs in Cxcl13-Cre *I133^{fl/fl}* and Cre-negative Ctrl mice (Fig. R3.3a and b), while intratumoral ILC2 cells were almost undetectable. Our data support the study cited by the reviewer (Moral JA Nature 2020 PMID 32076273) that shows that IL-33-mediated activation of tumor-infiltrating ILC2s and CD8⁺ T cells is crucial in orthotopic pancreatic tumors but not in heterotopic skin tumors. As a control, we analyzed the TDLN to demonstrate that we can reliably stain for ILCs and ILC2s and that their frequencies were not altered in Cxcl13-Cre *I133^{fl/fl}* compared to Ctrl mice (Fig. 3.3a, c and d).

Figure R3.3. FSC-specific IL-33 ablation does not impede intratumoral ILCs and ILC2s after artLCMV-TRP2 treatment. Mice were inoculated s.c. with B16F10 melanoma cells and treated on day 7 i.t. with artLCMV-TRP2. (a) Gating strategy for CD127⁺ CD25⁺ ILCs and CD127⁺ CD25⁺ ST2⁺ ILC2s in the tumor and the TDLN. (b to d) Frequency of CD127⁺ CD25⁺ ILCs in the tumor (b) and frequency of CD127⁺ CD25⁺ ILCs (c) and CD127⁺ CD25⁺ ST2⁺ ILC2s (d) in the TDLN in Cxcl13-Cre *Il33^{fl/fl}* and Cre-negative Ctrl mice. Pooled data from two independent experiments with n=6-8 mice.

We further characterized the immune cell composition in the tumor microenvironment of Cxcl13-Cre *Il33^{fl/fl}* and Cre-negative Ctrl mice treated i.t. with artLCMV-TRP2. On day 11 (day 4 after artLCMV-TRP2 injection), we found comparable frequencies of CD11b⁺ myeloid cells, CD11c^{hi} MHC II^{hi} DCs, CD11b⁺ Ly6C^{hi} inflammatory monocytes (IM) or F4/80⁺ MHC II⁺ tumor-associated macrophages (TAMs) as well as CD19⁺ B cells, CD8⁺ and CD4⁺ T cells and FoxP3⁺ CD4⁺ regulatory T cells (T_{regs}) in Cxcl13-Cre *Il33^{fl/fl}* and Ctrl mice (see Fig. R3.4a and c). However, analysis of the immune milieu on day 15 (day 8 after artLCMV-TRP2 treatment) unveiled a significant reduction of intratumoral CD8⁺ T cells in mice with *Il33*-deficiency in Cxcl13-Cre⁺ FSCs (new ED Fig. 7b-d and Fig. R3.4 a-c). These data support the finding that IL-33 signals through its receptor ST2 expressed on tumor-infiltrating CD8⁺ T cells (Fig. 6a) to preserve functional activation locally within the TME. Thus, IL-33 catered by Cxcl13-Cre⁺ FSCs in the TME can act directly on CD8⁺ T cells.

Please see also our response to point 4 of reviewer 4 showing that adoptive transfer of ST2-deficient T cells results in significantly reduced T cell effector function.

Figure R3.4. FSC-specific IL-33 ablation solely influences abundance of intratumoral CD8⁺ T cells. Mice were inoculated s.c. with B16F10 melanoma cells and treated on day 7 i.t. with artLCMV-TRP2. (a and b) Frequency of CD11b⁺ myeloid cells, CD11c^{hi} MHC II^{hi} DCs, CD11b⁺ Ly6C^{hi} inflammatory monocytes (IM) or F4/80⁺ MHC II⁺ tumor-associated macrophages (TAMs) as well as CD19⁺ B cells, CD8⁺ and CD4⁺ T cells on (a) day 11 (d4 after artLCMV-TRP2) and (b) day 15 (day 8 after artLCMV-TRP2). (c) Frequency of FoxP3⁺ CD4⁺ T cells. Pooled data from two independent experiments with n=5-10 mice.

9. The relevant recent IL-33 literature should be cited.

This comment has been addressed and we have included additional recent citations for IL-33 in antitumor immunity in the discussion.

10. *Cxcl13* is not fibroblast specific. It is expressed by many other cells, including T cells and dendritic cells. Therefore, crossing *Cxcl13-Cre/tdTom* mice to *Il33fl/fl*, to generate *Cxcl13-Cre Il33fl/fl* mice would result in IL-33 ablation in multiple cell types, and is not CAF specific. This limitation is completely ignored by the authors in their suggested mechanism. While fibroblast-specific Cre mice are problematic (and beyond the scope), this limitation and alternative interpretation of the results must be addressed.

We can assure the reviewer that we have carefully evaluated the targeting of the *Cxcl13-Cre* transgene in the skin (Fig. R3.5a) and the tumor microenvironment (Fig. R.3.5b) by flow cytometry. Within these tissues, we found EYFP⁺ cells in CD45⁻ EpCAM⁻ CD31⁻ PDPN⁺ cells. Thus, these genetic models are useful tools to elucidate phenotype and function of FSC subsets in different tissues during homeostasis and in inflammatory diseases and cancer. Moreover, fibroblast-specific Cre mouse models aid in investigating the role of interesting candidate genes in specific FSC populations. Thus, our data demonstrate that the *Cxcl13-Cre/tdTom* EYFP model is a useful tool to study microenvironmental niches build by *Cxcl13*-expressing iCAFs in the TME to sustain T cell functionality and thereby tumor control.

Figure R3.5. *Cxcl13-Cre*⁺ cells are labelled by EYFP and is fibroblast-specific in skin and tumors. (a and b) EYFP expression in *Cxcl13-Cre/tdTom EYFP* and *Cre- Ctrl* mice (gated on *Live Ter119*⁻ cells) in skin (a) and *B16F10* tumors (b). Gating scheme for fibroblasts from *Cxcl13-Cre/tdTom EYFP* mice found within *CD45*⁻*EpCAM*⁻ (b) and *CD31*⁻ cells (black) and for EYFP⁺ fibroblasts found within *Live Ter119*⁻ cells (green).

Reviewer #4 (Remarks to the Author): with expertise in cancer associated fibroblasts and scRNA-seq

This study by Ring et al presents some interesting data identifying a critical role for fibroblast reprogramming in sustained response to anti-melanoma vaccination. These data are novel and likely to be of significant interest to the Nature Communications readership. My main concern is a lack of detail regarding why/how the viral vector used elicits the fibroblast reprogramming reaction, which may limit the impact these findings could have and their potential application.

Major points:

1. The terminology used to describe the fibroblast subpopulations identified needs to be addressed.

- The markers used for the myoCAF2 designation are questionable and not consistent with the studies that originally defined the MyoCAF term. In Extended Data Figure 6 the markers shown for the MyoCAF2 population includes multiple genes upregulated following hypoxia, including Vegfa. This is consistent with the VEGF+ CAF subpopulation described by Grauel et al. (NCOMMS 2020). The authors should amend the terminology used to reflect this.*

We are glad that reviewer 4 considers that this work identifies a critical role for fibroblast reprogramming in the sustenance of antitumoral responses in tumor vaccination and that these data are novel. We agree with the reviewer that the terminology used to describe fibroblast subpopulation in tumors and other tissues requires particular attention. Indeed, the terminology and the fibroblast nomenclature, their origin and identity are critically discussed in the field Paper (Sahai et al. 2020 PMID 31980749). Our rationale of naming the cluster myCAF2 is based on the expression of genes characteristic of immune-suppressive myCAFs such as *Acta2*, *Col15a1*, *Tgfb1* and *Lrrc15* (Biffi et al. 2019 PMID 30366930, Elyada et al. 2019 PMID 31197017, Dominguez et al. 2020 PMID 31699795). However, we also found expression of *H2-Ab1* but did not term this cluster “antigen-presentating CAFs” (apCAFs) as other genes recently described for apCAFs were not found (Elyada et al. 2019 PMID 31197017). We agree with the reviewer that this cluster is similar to the VEGF+ CAF subpopulation described by Grauel et al and we reference this in the manuscript.

2. How does the intratumoural artLCMV-TRP2 treatment modify myoCAF features of the tumour microenvironment?

- The UMAP plot (Figure 3a) presented suggests a phenotypic shift in the myoCAF1 population. The authors should present data on Differential expression analysis between PBS and artLCMV-TRP2 treated myoCAF1, to determine whether the changes in gene expression are similar or different between iCAFs and myoCAFs. Given that myoCAFs have been shown to be the principal CAF subpopulation involved in immune checkpoint non-response, if these cells can also be reprogrammed to an IL-33+ phenotype this would significantly enhance the potential reach of the findings presented in this paper.*

To avoid excessive display of data in the manuscript and because we found a pronounced shift from iCAF1 to iCAF2 following artLCMV injection, we decided to focus on the role of iCAFs in sustaining CD8⁺ T cell function in the TME. As suggested by the reviewer, we determined DE gene expression by comparing iCAFs and myCAF1 and found that artLCMV injection elicits a shift of both CAF subsets towards an immune-stimulatory state (Fig. R4.1). However, the immune-supportive and -activating state was more pronounced in the *Cxcl13*-expressing iCAF cluster which is illustrated by the diffusion maps in Fig. 5e, f and ED Fig. 6c. Thus, we conclude that local artLCMV vector treatment induces reprogramming of the different subsets in the CAF landscape towards an immune-supportive phenotype.

Figure R4.1. Intratumoral artLCMV strongly activates iCAF subset with a pronounced immune-stimulatory gene signature. DE gene analysis of iCAFs and myCAF1 following i.t. artLCMV-TRP2 vs PBS control treatment.

3. Does *Cxcl13-IL-33*fl/fl impact the phenotype of stromal cells?

• In Figure 3 the authors show that artLCMV-TRP2 treatment causes fibroblasts to upregulate multiple genes that could be involved in the recruitment and activation of CD8⁺ T-cells. Figure 4 shows that intratumoural artLCMV-TRP2 treatment in *Cxcl13-IL-33*fl/fl mice fails to control tumour growth. The conclusions drawn suggest that this is due to the direct action of CD8⁺ T-cells by IL-33. However, an alternative explanation could be that artLCMV-TRP2 treatment induced IL-33 is responsible for the fibroblast reprogramming. Experiments should be performed to determine whether IL-33 acts directly through CD8 activation, indirectly by reprogramming fibroblasts or a combination of the two mechanisms.

Please see also our response to point 8 of reviewer 3 with the result that *Il33*-deficiency in *Cxcl13-Cre*⁺ FSCs solely reduced intratumoral CD8⁺ T cell abundance while other immune cell populations were not affected. In addition, we addressed this comment by analyzing the phenotype of tumor-associated FSCs in i.t. artLCMV-TRP2-treated *Cxcl13-Cre*/tdTom EYFP and *Cxcl13-Cre*/tdTom *Il33*^{fl/fl} EYFP mice. The abundance of PDPN⁺ FSCs, CD31⁺ BECs and PDPN⁻ CD31⁻ DN cells was not altered due to the *Il33*-deficiency in *Cxcl13-Cre*⁺ FSCs (Fig. R4.2a). The frequency of EYFP⁺ PDPN⁺ FSCs as well as the current expression of CXCL13 as assessed by tdTomato⁺ cells within the EYFP⁺ PDPN⁺ FSCs was comparable between both groups (Fig. R4.2b and c). Likewise, we did not observe differences in the proportion of iCAFs identified by expression of CD26 and CD34 or in expression levels of activation molecules such as MHC I (Fig. R4.2d and e). Thus, these data suggest that IL-33 is not acting in an autocrine manner on the tumor fibroblasts, but rather directly influences tumor-infiltrating T cells.

Figure R4.2. FSC-specific IL-33 does not alter the phenotype tumor-associated FSCs. *Cxcl13-Cre/tdTom EYFP* and *Cxcl13-Cre/tdTom Il33^{fl/fl} EYFP* mice were inoculated s.c. with B16F10 melanoma cells and treated on day 7 i.t. with artLCMV-TRP2. Stromal cells were analysed on day 11. (a) Frequencies of stromal cell subsets in the tumor. (b) Frequency of EYFP⁺ PDPN⁺ in *Cxcl13-Cre/tdTom EYFP* mice. (c) Frequency of tdTom⁺ cells among EYFP⁺ PDPN⁺ cells. (d) Frequency of iCAFs among EYFP⁺ PDPN⁺ identified by CD26 and CD34 cells. (e) Mean expression of MHC I (H2-K^b and D^b) on EYFP⁺ PDPN⁺ FSCs. Data from one to two independent experiments with n=5-10 mice.

Analysis of the tumor immune milieu revealed that intratumoral CD8⁺ T cells were significantly reduced in *Cxcl13-Cre Il33^{fl/fl}* compared with Ctrl mice, while other immune cell population were not affected (Fig. 6d, ED Fig. 7b-d and Fig. 3.4 b and c in Point 8 Reviewer 3). Moreover, tumor-infiltrating CD8⁺ T cells expressed the IL-33 receptor ST2 (Fig. 6a). We performed additional experiments to demonstrate that direct IL-33/ ST2 signaling on CD8⁺ T cells is crucial to foster functional T cell activation and prevent exhaustion. We adoptively transferred P14 CD8⁺ T cells (specific for the LCMV epitope GP₃₃₋₄₁) into tumor-bearing *Cxcl13-Cre Il33^{fl/fl}* or Ctrl mice and treated them i.t. with artLCMV-TRP2 (Fig. R4.3a). As shown in the manuscript for endogenous intratumoral TRP2-specific CD8⁺ T cells (Fig. 6g), expression of effector cytokines was significantly reduced in P14 CD8⁺ T cells in *Cxcl13-Cre Il33^{fl/fl}* mice (Fig. R4.3b) confirming that FSC-derived IL-33 is crucial to sustain CD8⁺ T cell functionality. Importantly, the production of IFN- γ and TNF- α was significantly reduced in transferred ST2-deficient compared to ST2-proficient P14 CD8⁺ T cells (Fig. R4.3c). In sum, our newly generated data provide evidence that IL-33 provided by *Cxcl13-Cre*⁺ FSCs acts directly on intratumoral ST2-expressing CD8⁺ T cells ensuring sustenance of T cell effector function and tumor rejection. We have included these data in the manuscript as ED Fig. 8.

Figure R4.3. Influence of ST2 signalling on intratumoral CD8⁺ T cell differentiation and functionality. (a) Mice were inoculated s.c. with B16F10 melanoma cells. On day 7, 2×10^5 P14 CD8⁺ T cells (specific for the LCMV GP₃₃₋₄₁ epitope) or ST2-deficient were adoptively transferred and mice were treated i.t. with artLCMV-TRP2. (b) Frequency of IFN- γ ⁺ TNF- α ⁺ tumor-infiltrating P14 CD8⁺ T cells in *Cxcl13-Cre Il33^{fl/fl}* or Ctrl mice following stimulation with the GP₃₃₋₄₁ peptide. (c) Frequency of IFN- γ - and TNF- α -producing ST2 (*Il1rl1*)-proficient or -deficient P14 CD8⁺ T cells transferred into B6 mice following stimulation with the GP₃₃₋₄₁ peptide. Pooled data from two independent experiments with n=8-10 mice.

4. Could intratumoural artLCMV-mediated fibroblast reprogramming also be used as an immunotherapy adjunct?

• Grauel et al. (NCOMMS 2020) showed that reprogramming the stroma using TGF-beta blockade generated a fibroblast subpopulation, similar to that described in this study following artLCMV-TRP2 treatment, which increased the efficacy of PD1 immunotherapy. Could combined artLCMV treatment and PD1 immunotherapy be an attractive strategy for patients without vaccine actionable mutations?

This is an important point as only a fraction of cancer patients respond to checkpoint inhibitors therapy. Resistance to PD-1 / PD-L1 therapy is correlated with the absence of intra- / peritumoral T cells. A growing number of preclinical studies and clinical trials is assessing combination treatments to therapeutically target immune-suppressive CAFs to enhance the efficacy of checkpoint inhibitor therapy. Our study shows that viral vector-mediated remodeling of the FSC landscape in the TME is crucial for successful tumor immunotherapy. While we have not observed a synergistic effect of combined i.t. artLCMV-TRP2 and anti-PD-1 treatment in the B16F10 model (Fig. R4.4), we have experimental evidence that only the combination treatment resulted in a complete tumor rejection in the MC38 tumor model. In sum, we agree that the striking plasticity of the tumor stromal compartment facilitates remodeling of fibroblasts by therapeutic interventional approaches (e.g. TGF-beta blockade or viral-based vectors) and thereby the generating of immune-supportive microenvironmental niches within the TME. Currently, artLCMV vectors are in clinical trials for the treatment of patients with human papilloma virus-induced head and neck cancer (ClinicalTrials.gov Identifier NCT0418021) and emerging data will show their potency in cancer therapy. Our study provides the rationale for local application of artLCMV vectors to generate immune-supportive niches for T cells in the TME and make tumors susceptible for checkpoint inhibitor therapy. Moreover, we have demonstrated translational relevance as tumor growth of contralaterally s.c. injected tumors or pulmonary metastasis is significantly reduced in mice administered i.t. with artLCMV-TRP2 into the accessible tumor.

Figure R4.4. Intratumoural artLCMV-TRP2 treatment does not show a synergistic effect with combined anti-PD-1 therapy. (a and b) Mice were injected s.c. with 2×10^5 B16F10 and treated either i.t. with artLCMV-TRP2. Mice received 200 ug anti-PD-1 i.p. twice per week starting on day 7. (a) Tumor kinetic and (b) survival of mice.

5. Is the fibroblast reprogramming described specific to the Trp2 antigen or artLCMV vectors?

• All the experiments analysing fibroblast phenotypes are compared to PBS controls. What happens to fibroblast phenotypes when transduced with another antigen or a different type of viral vector?

Our data show that tumor rejection after i.t. artLCMV-TRP2 treatment was dependent on expression of the TRP2 melanoma antigen by the tumor cells as well as by the vaccine vector (Fig. 1d and e). Please see also our responses to Point 3 of Reviewer 1, Point 1 of Reviewer 2 and Point 3 of Reviewer 3; these

data show that the artLCMV vectors induce the reprogramming of the tumor-associated FSCs.

Moreover, we have analyzed the phenotype of tumor-associated FSCs in mice treated with artLCMV-TRP2 or artLCMV-GFP (transduction with an irrelevant antigen). The frequency of PDPN⁺ FSCs, CD31⁺ BECs and PDPN⁻ CD31⁻ DN cells were increased after artLCMV treatment independent of the vector-delivered antigen (Fig. R4.3a). Likewise, activation of FSCs by LCMV-based vectors was comparable between artLCMV-TRP2 and artLCMV-GFP as shown by significant increase of MHC I molecules on PDPN⁺ FSCs (Fig. R4.3b).

Figure R4.3. artLCMV-induced reprogramming of tumor-associated FSCs is independent of the vaccine-delivered antigen. Mice were inoculated s.c. with B16F10 melanoma cells and treated on day 7 i.t. with artLCMV-TRP2 or artLCMV-GFP. Stromal cells were analyzed on day 11. (a) Frequencies of stromal cell subsets in the tumor. (b) Mean expression of MHC I (H2-K^b and D^b) on PDPN⁺ FSCs. Data from two independent experiments with n=4-10 mice.

Minor points:

There are some errors in the legend for figure 3. For example, Panel k is referred to when describing the statistical tests used but there is no panel k.

Thank you for the careful attention to the text and legends. We have amended the legend accordingly.

Reviewers' Comments:

Reviewer #1:

Remarks to the Author:

The authors have responded to my comments and the revised paper is much stronger. Nice work!

Howard Kaufman

Reviewer #2:

Remarks to the Author:

Ring et al addressed all points raised by the reviewers. All questions raised by myself are extensively answered and now the manuscript is an excellent piece of work, which will give a major contribution to the scientific field. I am sure that the mechanisms described in this manuscript will be of high relevance, not just for Arenavirus-based therapy, but also for virotherapy in general. I have no more points and suggest publication without any further delay.

Reviewer #3:

Remarks to the Author:

In the revised version the authors have addressed most of my comments, and the manuscript is much improved.

One remaining point to consider is that some of the explanations I suggested were addressed in detail in the rebuttal, but not in the text (e.g. CAF specificity of CXCL13 targeting).

Also, discussing the clinical limitations of the approach will provide the reader with a more complete picture for this potential therapeutic approach.

Reviewer #4:

Remarks to the Author:

I would like to thank the authors for their clear and comprehensive response to my review of the original manuscript. In my opinion this manuscript will have a valuable impact on the field, providing novel insight into possible mechanisms for reprogramming cancer associated fibroblasts in order to enhance anti-tumour immunity.

Point-by-point reply

We would like to thank the editors for the thorough work on our manuscript. All editorial requests have been addressed and corresponding changes are listed in the document:

NCOMMS-20-48930A_Extended_comments_1624869628_56.pdf

Comments to:

Reviewer #3

Remarks to the Author:

In the revised version the authors have addressed most of my comments, and the manuscript is much improved.

One remaining point to consider is that some of the explanations I suggested were addressed in detail in the rebuttal, but not in the text (e.g. CAF specificity of CXCL13 targeting).

Also, discussing the clinical limitations of the approach will provide the reader with a more complete picture for this potential therapeutic approach.

We now discuss both points mentioned by Reviewer 3 (underlined text in the Discussion section).